# DYNAMIC INTERFERENCE MODELING FOR ESTIMATING TREATMENT EFFECTS FROM DYNAMIC GRAPHS

## ABSTRACT

Estimating treatment effects can assist decision-making in various areas, such as commerce and medicine. One application of the treatment effect estimation is to predict the effect of an advertisement on the purchase result of a customer, known as individual treatment effect (ITE). In online websites, the outcome of an individual can be affected by treatments of other individuals, as people often propagate information with their friends, a phenomenon referred to as interference. Prior studies have attempted to model interference for accurate ITE estimation under a static network among individuals. However, the network usually changes over time in real-world applications due to complex social activities among individuals. For instance, an individual can follow another individual on one day and unfollow this individual afterward on an online social website. In this case, the outcomes of individuals can be interfered with not only by treatments for current neighbors but also by past information and treatments for past neighbors, which we refer to as *dynamic interference*. In this work, we model dynamic interference for the first time by developing an architecture to aggregate both the past information of individuals and their neighbors. Specifically, our proposed method contains a mechanism that summarizes historical information of individuals from previous time stamps, graph neural networks that propagate information about individuals within every time stamp, and a weighting mechanism that estimates the importance of different time stamps. Moreover, the model parameters should gradually change rather than drastically because information of every individual gradually changes over time. To take it into account, we also propose a variant of our method to evolve the model parameters over time with long short-term memory. In our experiments on multiple datasets with dynamic interference, our methods outperform existing methods for ITE estimation because they are unable to capture dynamic interference. This result corroborates the importance of dynamic interference modeling.

## 1 INTRODUCTION

Treatment effect estimation has been applied to decision-making in various areas, such as medicine (Ma et al., 2022a; Schnitzer, 2022; Chang et al., 2023) and commerce (Nabi et al., 2022; Ellickson et al., 2023; Waisman et al., 2024). For instance, estimating treatment effects can help business owners understand whether an advertisement encourages customers to purchase the advertised item. Then, owners can make informed decisions about whether to proceed with the promotion based on the treatment effect estimation results. The individual treatment effect (ITE) quantifies the relative change of an individual outcome with/without treatment.

Our goal is to estimate treatment effects from observational graph data under a dynamic environment. In this scenario, data typically contains covariates of individuals, a network among individuals, treatments, and outcomes, all of which change over time. This gives rise to dynamic ITE. Estimating dynamic ITE enables us to deeply understand how the effects of a treatment change over time for individuals, which can result in more reasonable decisions. In a graph, an individual outcome can be influenced by treatments assigned to its neighboring individuals, a phenomenon referred to as interference (Ma & Tresp, 2021). A dynamic network may result in more complex interference among individuals than a static network. For instance, a user can follow another user and unfollow this user afterward in an online social application. In this case, an individual possibly receives interference from his/her own past treatment, neighbors in the past time stamps, and even from individuals never

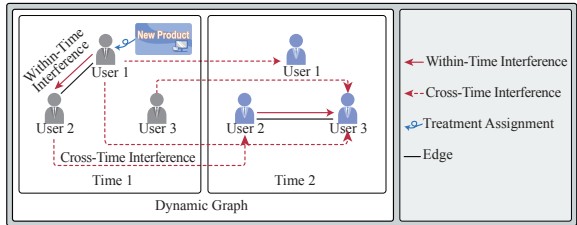

Figure 1: An example of dynamic interference between two time stamps. In this example, the advertisement is only assigned to User 1 at Time 1, but User 3 can be influenced at Time 2 because User 2 mediates Users 1 and 3 at different time stamps. Importantly, Users 1 and 3 are not neighboring at any time stamp, which introduces challenges in capturing dynamic interference.

neighboring at any time stamp. We illustrate an example in Figure 1, where the advertisement is only assigned to User 1 at Time 1. However, interference occurs not only between Users 1 and 2 at Time 1 but also between User 1 at Time 1 and User 3 at Time 2 via User 2. We refer to the former interference occurring within the same time stamp as *within-time interference* (in red and solid lines), and occurring across time stamps as *cross-time interference* (in red and dashed lines). At any past time, interference received by every individual from their neighbors can be carried over to different time stamps and propagated within these time stamps, which results in severe cross-time interference among individuals. The repeated occurrence of within-time and cross-time interference in the dynamic graph constitutes *dynamic interference*. Properly modeling dynamic interference is crucial; Otherwise, we end up with inaccurate ITE estimation, which results in unreasonable decisions.

To capture dynamic interference, we need to overcome the following challenges: (I) Historical information aggregation. Each time stamp often contributes interference differently. This introduces challenges for aggregating historical information. (II) Within-time interference modeling under a dynamic environment. At every time stamp, individuals can have different covariates (e.g., age), receive new treatments, and dynamic networks among individuals, all of which result in different within-time interference at different time stamps. (III) Cross-time interference modeling. As shown in Figure 1, cross-time interference can occur among Users 1 and 3 who are not neighboring at any time stamp, which introduces a significant challenge in capturing cross-time interference. Previous approaches fail to model dynamic interference, as they cannot overcome challenges (I), (II), and (III) jointly. Existing studies to ITE estimation capture within-time interference only (Ma & Tresp, 2021; Jiang et al., 2023), or model interference in panel data that have dynamic treatments only and thus are incapable of handling dynamic graphs (Wang, 2021), or estimate ITE from dynamic graphs but do not model interference among individuals (Ma et al., 2021).

To overcome these challenges, we propose a novel approach: **D**ynamic **I**nterference modeling for estimating **T**reatment **E**ffects from dynamic graph (DITE). The key idea of our approach is to model the propagation of dynamic interference across time and individuals. To this end, we design a dynamic interference modeling (DIM) layer. Specifically, every DIM layer contains a mechanism that aggregates historical information. To overcome the challenge (I), DIM first uses a weighting mechanism to estimate the importance of different time stamps, inspired by attention mechanism (Vaswani et al., 2017). Subsequently, the DIM layer summarizes historical information weighted by the estimated importance of different time stamps. Next, DIM applies single-layered GNNs (Welling & Kipf, 2016) to propagate information among individuals within every time stamp, which captures within-time interference at every time stamp and overcome the challenge (II). By stacking DIM layers, we can overcome the challenge (III) and capture dynamic interference received by every individual. Moreover, as the covariates and treatments usually change gradually over time, model parameters should also change gradually over time. To take it into account, we propose a variant of DITE, which uses long short-term memory (LSTM) (Hochreiter & Schmidhuber, 1997) to evolve the model parameters of DITE. Results on extensive experiments in Section 5 reveal the powerful ability of the proposed methods in estimating ITE with dynamic interference.

The contributions of this study can be summarized as follows:

- We formalize the problem of ITE estimation with dynamic interference, which is a novel and challenging issue.

- This study proposes methods to model dynamic interference and estimate treatment effects with dynamic interference.
- Results of extensive experiments reveal that our methods outperform existing methods in ITE estimation with dynamic interference, which suggests the importance of modeling dynamic interference.

## 2 RELATED WORK

**ITE estimation without interference.** Many existing methods for ITE estimation, such as balancing neural network (BNN) (Johansson et al., 2016), deep treatment-adaptive architecture (DTANet) (Li et al., 2022), and counterfactual regression (CFR) (Shalit et al., 2017) assume that there is no interference among individuals. This assumption hardly holds for real-world data, as individuals in real-world data are usually connected, such as Flicker (Wang et al., 2013) and BlogCatalog datasets (Li et al., 2015), and often propagate information with neighboring individuals. Some studies consider estimating ITE from graph data (Guo et al., 2020; Chu et al., 2021; Ma et al., 2021). However, these methods make use of graphs to discover latent information but still do not take interference into account. For a more thorough literature review on this line, confer the survey paper (Yao et al., 2021).

**ITE estimation with interference under static environments.** The earliest studies introduced *group-level* interference (Hudgens & Halloran, 2008; Tchetgen & VanderWeele, 2012; Liu & Hudgens, 2014). They split individuals into several subgroups and consider interference only within subgroups. Group-level interference may not apply to real-world situations, as interference can occur among individuals in different subgroups. The follow-up studies model interference, by assuming that interference exists among some close neighbors (Aronow & Samii, 2017; Rakesh et al., 2018; Viviano, 2019; Forastiere et al., 2021; Jiang & Sun, 2022). In real-world data, interference can propagate widely over a graph not only close neighbors. To address this issue, *networked* interference (Ma & Tresp, 2021; Forastiere et al., 2022; Huang et al., 2023; Sui et al., 2024) is assumed to consider interference propagation over an entire graph. GNN-based estimators (Ma & Tresp, 2021) were proposed to capture networked interference through the propagation mechanism of GNNs (Welling & Kipf, 2016). As the structure of a graph may be hidden due to privacy preservation (Sävje et al., 2021; Cortez et al., 2022), a subsequent study (Bhattacharya et al., 2020) construct graph structures for interference modeling and treatment effect estimation. To estimate ITE from more convoluted graphs, HyperSci (Ma et al., 2022b) is proposed to capture joint interference of multiple individuals to an individual on hypergraphs using a hypergraph-GCN method (Bai et al., 2021), and HINITE (Lin et al., 2023) is proposed to model interference propagation on heterogeneous graphs. However, these methods model within-time interference under a static environment only.

**ITE estimation under dynamic environments.** Wang (2021) considers interference from past time in panel data that have dynamic treatments only, whereas dynamic graph data has dynamic covariates of individuals and a dynamic topological structure among individuals. (Zhu et al., 2024) considers dynamic networks only, but static covariates and treatments, whereas we focus on dynamic graphs where both covariates and treatments are dynamic. DNDC (Ma et al., 2021) and NEAT (Ma et al., 2023) estimate ITE from a dynamic graph but does not model interference. CF-GODE (Jiang et al., 2023) considers interference under a dynamic environment and assumes that the treatment and covariates of individuals can change, whereas the graph structure of individuals remains unchanged over time. Moreover, CF-GODE assumes that interference exists among immediate neighbors at the same time stamp, i.e., one-hop within-time interference only. However, real-world data can often change its graph structure as seen in PeerRead dataset (Kang et al., 2018), and there often exists cross-time and dynamic interference due to complex social activities, as shown in Figure 1. To overcome this issue, our work relaxes the restriction on graph structures and models networked interference under a dynamic environment that contains dynamic covariates, treatments, and networks.

## 3 PROBLEM SETTING

We aim to estimate treatment effects from observational graphs under a dynamic environment. Let $t$ be a time stamp. We use $\boldsymbol{x}_i^t \in \mathbb{R}^d$, $\tau_i^t \in \{0, 1\}$, and $y_i^t \in \mathbb{R}$ to denote the covariates, treatment, and observed outcome of an individual $i$ at the time stamp $t$, respectively. Let $\mathbf{X}^t = \{\boldsymbol{x}_i^t\}_{i=1}^n$, $\mathbf{T}^t = \{\tau_i^t\}_{i=1}^n$, and $\mathbf{Y}^t = \{y_i^t\}_{i=1}^n$ be the set of covariates, treatments, and observed outcomes at the

time stamp $t$, respectively, where $n$ is the number of individuals. We use non-bold, italicized, and capitalized letters (e.g., $X_i^t$) to denote random variables, subscript $-i$ (e.g., $\mathbf{X}_{-i}^t$) to denote all other individuals except $i$, and superscript $\leq t$ (e.g., $\mathbf{X}_i^{\leq t} = \{\boldsymbol{x}_i^{t_0}\}_{t_0=1}^t$ and $\mathbf{X}^{\leq t} = \{\mathbf{X}^{t_0}\}_{t_0=1}^t$) to denote time stamps up to time stamp $t$. An individual with $\tau^t = 1$ is treated, and $\tau^t = 0$ is controlled at $t$.

**Dynamaic graphs.** We use an adjacency matrix $\boldsymbol{A}^t \in \{0,1\}^{n \times n}$ to represent the structure of a dynamic graph at the time stamp $t$. For simplicity, we assume the graph is undirected and unweighted, but it can be naturally extended to directed and weighted graphs. If there is an edge between individuals $i$ and $j$ at the time stamp $t$, $A_{ij}^t = A_{ji}^t = 1$; Otherwise, $A_{ij}^t = A_{ji}^t = 0$. Let $A_{ii}^t = 0$, $\mathbf{A}^{\leq t} = \{\boldsymbol{A}^{t_0}\}_{t_0=1}^t$, and $\mathbf{N}_i^t$ be the set of neighbors of $i$ at the time stamp $t$.

**ITE estimation with dynamic interference.** Dynamic graph data can be denoted by $\{\mathbf{X}^t, \mathbf{T}^t, \mathbf{Y}^t, \boldsymbol{A}^t\}_{t=1}^{t_{\max}}$, where $t_{\max}$ is the number of all time stamps. Let $\mathbf{H}^t = \{\mathbf{X}^{\leq t-1}, \mathbf{T}^{\leq t-1}, \mathbf{A}^{\leq t-1}\}$. We assume that there exists interference among individuals in dynamic graphs. In this case, the outcome of an individual is not only influenced by its own treatments and co-variates but also by those of its neighbors (Rakesh et al., 2018; Ma & Tresp, 2021). In dynamic graphs, individuals can receive interference from current neighbors and their past covariates and treatments. At any past time $t'$ (where $t' < t$), interference received by every individual from their neighbors can be carried over to the current time $t$, and propagated within the current time $t$, which results in severe cross-time interference among individuals. The repeated occurrence of within-time and cross-time interference in the dynamic graph constitutes dynamic interference. Importantly, an individual cannot receive interference from future time stamps. To formalize the ITE with dynamic interference, we use $\boldsymbol{s}_i^t \in \mathbb{R}^{d'}$ to represent summary information of dynamic interference that the individual $i$ may receive from $\mathbf{X}^{\leq t}$ and $\mathbf{T}^{\leq t}$ on dynamic graphs $\mathbf{A}^{\leq t}$. We call $\boldsymbol{s}_i^t$ interference representation. Specifically, we assume that $\boldsymbol{s}_i^t$ can be captured by some aggregation function $\mathrm{AGG}^t(\mathbf{X}_{-i}^t, \mathbf{T}_{-i}^t, \mathbf{H}^t, \boldsymbol{A}^t) = \boldsymbol{s}_i^t$ for every time stamp $t$. The potential outcomes of the individual $i$ in dynamic graphs, denoted by $Y_i^t(1, \boldsymbol{s}_i^t)$ and $Y_i^t(0, \boldsymbol{s}_i^t)$, are real outcomes for the individual $i$ under $\boldsymbol{s}_i^t$ with treatment value $\tau_i^t = 1$ and $\tau_i^t = 0$, respectively. Then, we define the ITE with dynamic interference as follows:

$$\Delta_i^t = \mathbb{E}[Y_i^t(T_i^t = 1, S_i^t = \boldsymbol{s}_i^t) - Y_i^t(T_i^t = 0, S_i^t = \boldsymbol{s}_i^t) \mid X_i^t = \boldsymbol{x}_i^t]. \tag{1}$$

**Confounder.** Confounders are a part of covariates, which can simultaneously affect the treatment assignment and outcome (Yao et al., 2021), resulting in confounding bias to ITE estimation. The existence of confounders is a well-known issue when estimating the ITE from observational data (Shalit et al., 2017). For instance, we consider that treatment is whether a customer receives an advertisement. Young customers have more chances to see an advertisement. Meanwhile, young customers often prefer shopping more than elderly customers. In this case, age is a confounder. We address confounders in our proposal independently from capturing dynamic interference.

**Identifiability of ITE.** Subsequently, we discuss that ITE is identifiable with dynamic interference under a set of assumptions. First, we extend the neighbor interference assumption (Forastiere et al., 2021) to networked interference on dynamic graphs, as follows:

**Assumption 1** *For* $\forall i$, $\forall \mathbf{X}_{-i}^t, \tilde{\mathbf{X}}_{-i}^t, \forall \mathbf{T}_{-i}^t, \tilde{\mathbf{T}}_{-i}^t$, $\forall \boldsymbol{A}^t, \tilde{\boldsymbol{A}}^t$, *and* $\forall \mathbf{H}^t, \tilde{\mathbf{H}}^t$, *when* $\boldsymbol{s}_i^t = \mathrm{AGG}^t(\mathbf{X}_{-i}^t, \mathbf{T}_{-i}^t, \mathbf{H}^t, \boldsymbol{A}^t) = \mathrm{AGG}^t(\tilde{\mathbf{X}}_{-i}^t, \tilde{\mathbf{T}}_{-i}^t, \tilde{\mathbf{H}}^t, \tilde{\boldsymbol{A}}^t) = \tilde{\boldsymbol{s}}_i^t$, $Y_i^t(T_i^t = \tau_i^t, S_i^t = \boldsymbol{s}_i^t) = Y_i^t(T_i^t = \tau_i^t, S_i^t = \tilde{\boldsymbol{s}}_i^t)$ *hold.*

This assumption means that the outcome of an individual can receive interference from other individuals on the dynamic graph through $\boldsymbol{s}_i^t$, which is generated by $\mathrm{AGG}^t(\cdot)$. Next, we extend the consistency assumption (Forastiere et al., 2021) to dynamic interference:

**Assumption 2** $Y_i^t = Y_i^t(T_i^t = \tau_i^t, S_i^t = \boldsymbol{s}_i^t)$ *for the individual* $i$ *with* $\tau_i^t$ *and* $\boldsymbol{s}_i^t$.

This assumption means that the potential outcome is equal to the observed outcome under the same $\tau_i^t$ and $\boldsymbol{s}_i^t$ at $t$. Lastly, for simplicity, we adopt a strong assumption (i.e., unconfoundedness) widely used in existing works for interference modeling (Ma et al., 2022b; Lin et al., 2023; Sui et al., 2024):

**Assumption 3** *For any individual* $i$, *given the covariates, the treatment assignment and output of the aggregation function are independent of potential outcomes, i.e.,* $T_i^t, S_i^t \perp\!\!\!\perp Y_i^t(1, \boldsymbol{s}_i^t), Y_i^t(0, \boldsymbol{s}_i^t) | X_i^t$.

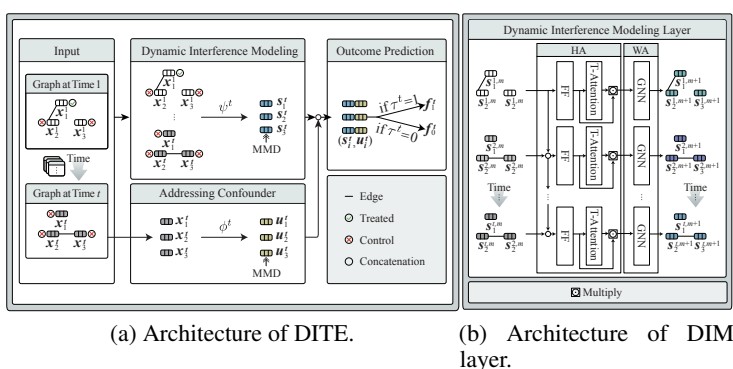

(a) Architecture of DITE.

(b) Architecture of DIM layer.

Figure 2: Architecture of the proposed DITE and DIM layers.

This assumption says that we can observe every feature that describes the difference between the treatment and the control group (Guo et al., 2020). Under the above assumptions, the expected potential outcomes $Y_i^t(\tau^t, \boldsymbol{s}_i^t)$ ($\tau^t = 1$ or $\tau^t = 0$) are identifiable, which can be derived as follows:

$$\mathbb{E}[Y_i^t \mid X^{\leq t} = \mathbf{X}^{\leq t}, T^{\leq t} = \mathbf{T}^{\leq t}, A^{\leq t} = \mathbf{A}^{\leq t}]$$

$$= \mathbb{E}[Y_i^t \mid X_i^t = \boldsymbol{x}_i^t, T_i^t = \tau^t, X_{-i}^t = \mathbf{X}_{-i}^t, T_{-i}^t = \mathbf{T}_{-i}^t, H^t = \mathbf{H}^t, A^t = \boldsymbol{A}^t] \quad \text{(Definition of } \mathbf{H}^t\text{)}$$

$$= \mathbb{E}[Y_i^t \mid X_i^t = \boldsymbol{x}_i^t, T_i^t = \tau^t, S_i^t = \boldsymbol{s}_i^t] \quad \text{(Aggregation function)}$$

$$= \mathbb{E}[Y_i^t(\tau^t, \boldsymbol{s}_i^t) \mid X_i^t = \boldsymbol{x}_i^t, T_i^t = \tau^t, S_i^t = \boldsymbol{s}_i^t] \quad \text{(Assumptions 1 and 2)}$$

$$= \mathbb{E}[Y_i^t(\tau^t, \boldsymbol{s}_i^t) \mid X_i^t = \boldsymbol{x}_i^t] \quad \text{(Assumption 3)}$$

The above proof suggests that once we accurately aggregate information on dynamic graphs into $\boldsymbol{s}_i^t$, we can identify $Y_i^t(1, \boldsymbol{s}_i^t)$ and $Y_i^t(0, \boldsymbol{s}_i^t)$. We discuss that ITE is identifiable in Appdendix A.

## 4 PROPOSED METHOD

Given dynamic graph data $\{\mathbf{X}^t, \mathbf{T}^t, \mathbf{Y}^t, \boldsymbol{A}^t\}_{t=1}^{t_{\max}}$, we aim to estimate treatment effects with dynamic interference. To this end, we propose DITE. Figure 2a shows the architecture of DITE with three individuals. DITE consists of three components. Specifically, the first component (explained in Section 4.1) models dynamic interference by propagating information of interference across individuals and times, and generates representations of individuals, which are referred to as interference representations. The second component (explained in Section 4.2) addresses confounders by learning balanced representations of covariates at every time stamp. The last component (explained in Section 4.3) consists of outcome predictors that infer potential outcomes using representations of interference and covariates. Moreover, DITE jointly trains $t_{\max}$ sub-networks containing these three components to capture dynamic changes in information. Subsequently, we present details for these components of a sub-network at a time stamp $t$. We will release codes of the proposed methods during the camera-ready phase.

### 4.1 DYNAMIC INTERFERENCE MODELING

To capture dynamic interference, it is essential to properly model the propagation of within-time and cross-time interference. To this end, we propose a dynamic interference modeling (DIM) layer, as illustrated in Figure 2b. A DIM layer contains a historical aggregation (HA) mechanism that aggregates all historical interference representations for every individual and generates new representations of individuals. As each time stamp contributes interference and outcome differently, the HA mechanism first applies a mechanism that automatically estimates the importance. This mechanism is inspired by the attention mechanism (Vaswani et al., 2017; Wang et al., 2019), so we refer to it as time-attention (abbreviated as T-Attention) mechanism. Subsequently, HA mechanism aggregates historical information by summarizing historical interference representations weighted by the learned importance. Next, the DIM applies a within-time aggregation (WA) mechanism that aggregates information for every individual within every time stamp using $t$ single-layered GNNs (Welling & Kipf, 2016), which take the outputs of the HA mechanism as inputs and generate

new interference representations. Therefore, by using a single DIM layer, we can capture within-time and cross-time interference from one-hop neighbors across time. By stacking multiple DIM layers, we can capture within-time and cross-time interference from multi-hops neighbors across time. For example, stacking two DIM layers can capture the cross-time interference from User 1 to User 3 in Figure 1. This enables the model to capture dynamic interference.

Specifically, given covariates $\mathbf{X}^{\leq t}$, treatments $\mathbf{T}^{\leq t}$, and dynamic networks $\mathbf{A}^{\leq t}$ up to $t$, we aim to capture dynamic interference and generate interference representations $\mathbf{S}^t$ of all individuals at every time stamp using a map function $\psi^t$. Every $\psi^t$ is achieved by stacking multiple DIM layers and output $\mathbf{S}^t$ at a time stamp $t$, i.e., $\mathbf{S}^t = \psi^t(\mathbf{X}^{\leq t}, \mathbf{T}^{\leq t}, \mathbf{A}^{\leq t})$. Let $M_\psi$ denote the number of layers for $\psi^t$. For an individual $i$, $\boldsymbol{s}_i^t$ is supposed to capture dynamic interference from other individuals on the dynamic graph up to the current time stamp ($\leq t$). Let $\boldsymbol{s}_i^{t,m}$ be an interference representation of the individual $i$ at time stamp $t$ and an input of the $m$-th DIM layer, $\boldsymbol{s}_i^{t,m+1}$ be a new interference representation for the individual $i$ generated by the current DIM layer, $\boldsymbol{p}_i^{t,m}$ be the output of a feed-forward (FF) layer of the T-Attention mechanism of an individual at time stamp $t$ of the $m$-th DIM layer, $\tilde{\boldsymbol{p}}_i^{t,m}$ be the output of HA mechanism of an individual at $t$ of $m$-th DIM layer, and $a_i^t$ be the inferred importance of the time stamp $t$. Let $\boldsymbol{b}$, $\boldsymbol{W}_{\mathrm{a}}^m$, and $\boldsymbol{q}$ be learnable parameters of the T-Attention mechanism and shared by all time stamps. For the first DIM layer, $\boldsymbol{s}_i^{t,0}$ is the concatenation of $\boldsymbol{x}_i^t$ and $\tau_i^t$. We add a self-loop for every individual to retain its information for propagation, i.e., $\tilde{\mathbf{N}}_i^t = \mathbf{N}_i^t \bigcup \{i\}$.

Now, we describe the architecture of the DIM layer in detail. First, the DIM layer performs the HA mechanism that applies the T-Attention mechanism to estimate the importance of every time stamp. The importance $a_i^t$ at a time stamp $t$ is calculated as follows:

$$\boldsymbol{p}_i^{t,m} = \boldsymbol{W}_{\mathrm{a}}^m \boldsymbol{s}^{t,m}, \quad w_i^t = \frac{1}{n}\sum_{i=1}^n \boldsymbol{q}^\top \sigma_{\mathrm{a}}(\boldsymbol{p}_i^{t,m} + \boldsymbol{b}), \quad a_i^t = \frac{\exp\left(w_i^t\right)}{\sum_{t_0=1}^t \exp\left(w_i^{t_0}\right)}, \tag{2}$$

where $\sigma_{\mathrm{a}}$ is an activation function. We use LeakyReLU (Maas et al., 2013) for $\sigma_a$ in our implementation. Next, HA mechanism aggregates historical information by using estimated importance, as follows:

$$\tilde{\boldsymbol{p}}_i^{t,m} = \sum_{t_0=1}^t a_i^{t_0} \boldsymbol{p}_i^{t_0,m}. \tag{3}$$

Finally, DIM layer uses WA mechanism to aggregate information within every time stamp. The WA mechanism at time stamp $t$ is performed as follows:

$$\boldsymbol{s}_i^{t,m+1} = \sigma\left(\frac{1}{|\tilde{\mathbf{N}}_i^t|}\sum_{j\in\tilde{\mathbf{N}}_i^t} \boldsymbol{W}_{\mathrm{gnn}}^{t,m}\tilde{\boldsymbol{p}}_j^{t,m}\right), \tag{4}$$

where $\sigma$ is an activation function, and $\boldsymbol{W}_{\mathrm{gnn}}^{t,m}$ be a learnable parameter matrix for the WA mechanism.

**Representation balancing.** An imbalance may exist in interference (Jiang et al., 2023), resulting in additional bias in ITE estimation. Consider again the example for confounders in Section 3, young customers typically receive advertisements (be treated) and have more young friends, which usually have higher rates of both purchase and seeing advertisements (high level of interference), whereas elderly customers have more elderly friends. To mitigate the imbalance, we add a discrepancy penalty to our loss function. A common choice of a discrepancy is a maximum mean discrepancy (MMD) (Shalit et al., 2017). Specifically, we minimize MMD between the representation distributions of interference in control and treated groups to mitigate imbalance in interference. Let $\mathrm{MMD}_\psi^t$ denote the estimated MMD of interference representations in different treatment groups at $t$.

**DITE with sliding window.** Handling numerous time stamps can be time-consuming and requires a significant amount of GPU memory when aggregating information from all past time stamps ($\leq t$). To mitigate this challenge, we propose a sliding window approach, which allows each individual to aggregate information from only the closest $K < t_{\max}$ time stamps, as information closest to the current time stamp is often the most important. For example, individuals usually have a better memory of recent events. The sliding window is applied to the HA mechanism, as follows:

$$a_i^t = \frac{\exp\left(w_i^t\right)}{\sum_{t_0=t-K}^t \exp\left(w_i^{t_0}\right)}, \quad \tilde{\boldsymbol{p}}_i^{t,m} = \sum_{t_0=t-K}^t a_i^{t_0}\boldsymbol{W}_a\boldsymbol{p}_i^{t_0,m}. \tag{5}$$

A larger value of $K$ allows the model to aggregate more information from the past data, but it increases the cost of GPU memory and training time. We recommend setting $K$ as large as possible when sufficient GPU resources and training time are available.

## 4.2 ADDRESSING CONFOUNDERS

To mitigate the confounding bias, we use an existing method (Shalit et al., 2017) that maps the covariates into a representation space where the distributions of representation of covariates in different groups are aligned well (Johansson et al., 2016; Shalit et al., 2017). To this end, we also minimize MMD between representation distributions of covariates in control and treated groups at every time stamp.

To be specific, $\phi^t$ is a feed-forward network at a time stamp $t$, which takes covariates $\mathbf{X}^t$ as input and outputs representations of covariates $\mathbf{U}^t = \phi^t(\mathbf{X}^t)$. Let $M_\phi$ be the number of layers of $\phi^t$ for every time stamp, and $\boldsymbol{u}_i^t$ be the representation of the covariates of the individual $i$ at the time stamp $t$. Similar to representation balancing for interference representation, we minimize MMD between the representation distributions of different treatment groups to mitigate confounding bias (Shalit et al., 2017). Let $\mathrm{MMD}_\phi^t$ be estimated MMD of covariate representations in different treatment groups at $t$.

## 4.3 OUTCOME PREDICTION

Given representations of interference $\mathbf{S}^t$, representation of covariates $\mathbf{U}^t$, and treatments $\mathbf{T}^t$ at time stamp $t$, we train two predictors that consist of multiple feed-forward layers to infer the outcomes with treatment $\tau^t = 1$ and $\tau^t = 0$ for every time stamp $t$.

Specifically, $f_1^t$ and $f_0^t$ are two feed-forward networks, and denote the predictor for treatment $\tau^t = 1$ and $\tau^t = 0$ at a time stamp $t$, respectively. They take $\boldsymbol{s}_i^t$ and $\boldsymbol{u}_i^t$ as input to predict outcomes. During training, they are optimized by minimizing the mean square error (MSE) between the predicted outcomes and the observed outcomes. Let $M_f$ denote the number of layers for predictors at every time stamp, $n_{\mathrm{train}}$ denote the size of the training set. The loss function of DITE at a time stamp $t$ consists of the MSE loss for outcome predictors and the sum of $\mathrm{MMD}_\psi^t$ and $\mathrm{MMD}_\phi^t$, where each term is traded off by a hyperparameter $\alpha$. To train $t_{\mathrm{max}}$ sub-networks of DITE jointly, we minimize the following loss function, which is averaged over all time stamps:

$$\mathcal{L} = \frac{1}{n_{\mathrm{train}}} \frac{1}{t_{\mathrm{max}}} \sum_{i=1}^{n_{\mathrm{train}}} \sum_{t=1}^{t_{\mathrm{max}}} \left( f_{\tau^t}^t(\boldsymbol{s}_i^t, \boldsymbol{u}_i^t) - y_i^t \right)^2 + \frac{\alpha}{t_{\mathrm{max}}} \sum_{t=1}^{t_{\mathrm{max}}} (\mathrm{MMD}_\phi^t + \mathrm{MMD}_\psi^t). \tag{6}$$

After training, we use $\phi^t$ and $\psi^t$ to generate representations of covariates and interference, respectively. To estimate ITE at the time stamp $t$, we design an ITE estimator: $\hat{\Delta}_i^t = f_1^t(\boldsymbol{s}_i^t, \boldsymbol{u}_i^t) - f_0^t(\boldsymbol{s}_i^t, \boldsymbol{u}_i^t)$, where the trained predictors $f_1^t$ and $f_0^t$ are used to predict the potential outcomes with $\tau^t = 1$ and $\tau^t = 0$, respectively.

## 4.4 VARIANT: DITE WITH WEIGHT EVOLUTION

As information (e.g., age) of individuals usually changes gradually over time, we need to prevent parameters from drastically changing over time. Therefore, we propose a variant that can automatically evolve the weights of every component of DITE. This variant is denoted as $\mathrm{DITE_{ev}}$.

We use LSTMs (Hochreiter & Schmidhuber, 1997) to evolve weights of every layer of DITE, inspired by the existing technology (Pareja et al., 2020). Every LSTM takes the weights of a layer at the last time as inputs and generates the weights of a layer at the current time, as shown in Figure 6. Specifically, let $\boldsymbol{W}_\phi^{t,m}$ and $\boldsymbol{W}_{f_{1/0}}^{t,m}$ be the weights of the $m$-th layer of the $\phi^t$ and $f_{1/0}^t$ at the time stamp $t$, respectively. The evolution of weights is modeled as follows:

$$\boldsymbol{W}_\phi^{t+1,m} = \mathrm{LSTM}(\boldsymbol{W}_\phi^{t,m}), \quad \boldsymbol{W}_{\mathrm{gnn}}^{t+1,m} = \mathrm{LSTM}(\boldsymbol{W}_{\mathrm{gnn}}^{t,m}), \quad \boldsymbol{W}_{f_{1/0}}^{t+1,m} = \mathrm{LSTM}(\boldsymbol{W}_{f_{1/0}}^{t,m}).$$

The details of LSTM are introduced in literature (Hochreiter & Schmidhuber, 1997; Pareja et al., 2020). Importantly, the difference in DITE and $\mathrm{DITE_{ev}}$ is that DITE first initializes parameters for $t_{\mathrm{max}}$ sub-networks and then directly trains $t_{\mathrm{max}}$ sub-networks, whereas $\mathrm{DITE_{ev}}$ only initializes and trains the sub-network at the first time stamp and uses LSTMs to evolve weights of every component

over time. This allows DITE$_{\text{ev}}$ to adaptively adjust its parameters based on the evolving dynamics of the data. Here, LSTMs and the sub-network of DITE$_{\text{ev}}$ at the first time stamp are training jointly.

## 5 EXPERIMENTS

In this section, we conducted experiments to answer the following research questions (RQ): **RQ 1**: How do the proposed methods perform with dynamic interference compared with baseline methods? **RQ 2**: Are DIM layers and information of dynamic graphs important to our methods? **RQ 3**: How sensitive to the trade-off parameter $\alpha$ are the proposed methods? **RQ 4**: How does the time horizon $t_{max}$ affect the performance of the proposed methods?

### 5.1 EXPERIMENTAL SETTING

We simulated outcomes with dynamic interference because the ground truth of ITE is hard to obtain due to the lack of counterfactual outcomes, i.e., the potential outcome for treatment $1 - \tau_i^t$. Many studies address this issue by simulating outcomes (Ma & Tresp, 2021; Ma et al., 2022b). We simulated outcomes with dynamic interference using three widely used datasets in the causal area (Guo et al., 2020; Ma et al., 2021; Jiang et al., 2023): Flickr (Wang et al., 2013), BlogCatalog (Li et al., 2015) (abbreviated as Blog), and PeerRead datasets (Kang et al., 2018).

**Outcome simulation.** We simulated outcomes with dynamic interference for all datasets as follows:

$$y_i^t = f_0(\boldsymbol{x}_i^t) + f_\tau(\tau_i^t, \boldsymbol{x}_i^t) + f_s(\mathbf{T}^{\leq t}, \mathbf{X}^{\leq t}, \mathbf{N}_i^{\leq t}) + \epsilon_i, \tag{7}$$

where $f_0(\boldsymbol{x}_i^t) = \boldsymbol{w}_0^\top \boldsymbol{x}_i^t$ simulates the outcome of an individual $i$ under treatment $\tau_i^t = 0$ without interference at time stamp $t$, and every element of $\boldsymbol{w}_0$ follows the Gaussian distribution $\mathcal{N}(0,1)$. The second function $f_\tau(\tau_i^t, \boldsymbol{x}_i^t) = \tau_i^t \cdot \boldsymbol{w}_1^\top \boldsymbol{x}_i^t$ simulates the ITE of the individual $i$, where $\boldsymbol{w}_1 \sim \mathcal{N}(0, \mathbf{I})$. We simulate the effect caused by dynamic interference through $f_s(\mathbf{T}^{\leq t}, \mathbf{X}^{\leq t}, \mathbf{N}_i^{\leq t}) = \sum_{c=1}^2 g_i^{t,c}$, where $g_i^{t,c} = f_{\text{Agg}}(\mathbf{G}^{\leq t, c-1}, \mathbf{N}_i^{\leq t}) = \frac{1}{|\mathbf{N}_i^t|} \sum_{j \in \mathbf{N}_i^t} f_h(\mathbf{G}_j^{\leq t, c-1})$, and $f_h(\mathbf{G}_j^{\leq t, c-1}) = \sum_{t_0=1}^t \boldsymbol{w}_j^{t_0,t} g_j^{t-t_0, c-1}$. Here, $\mathbf{G}^{\leq t, c}$ represents the calculated effects of dynamic interference of all individuals up to time stamp $t$ by $c$-th execution of aggregation function $f_{\text{Agg}}$, $g_i^{t,0} = \boldsymbol{w}_g^\top \text{Concat}(\boldsymbol{x}_i^t, \tau_i^t)$, $\boldsymbol{w}_i^{t_0,t} \sim \mathcal{N}(1 - t_0/t, (\frac{1}{t_{\max}})^2)$ controls the contribution of interference from time stamp $t_0$, and $\boldsymbol{w}_g$ follows the Gaussian distribution $\mathcal{N}(0, \mathbf{I})$. For simulating a dynamic environment, every element of $\boldsymbol{w}_0$, $\boldsymbol{w}_1$, and $\boldsymbol{w}_s$ can change over time by adding a random noise $\epsilon_{dy}$, which follows the Gaussian distribution $\mathcal{N}(0, 0.1)$. Lastly, $\epsilon_i \sim \mathcal{N}(0, 1)$ is a measurement noise. In Appendix D, we provide the illustration of the outcome simulation for Equation 7. In Appendix H, we show a simulation result with a different outcome simulation from what is presented here, corroborating the robustness of the proposed DIM layer to a specific form of the outcome simulation.

**Dataset description.** Flickr dataset (Wang et al., 2013) contains 7,575 users with more than 200,000 edges. BlogCatalog (abbreviated as Blog) dataset (Li et al., 2015) includes 5,196 users with more than 150,000 edges. We used covariates provided by literature (Guo et al., 2020) as the covariates at the first time stamp for the Flickr and Blog datasets. To simulate dynamic changes in the graph, we randomly add/remove edges with the probability of 5%, and perturb covariates at each time stamp by using a program provided by literature (Ma et al., 2021). We simulated $t_{\max} = 5$ for the Flickr and Blog datasets by default. PeerRead dataset (Kang et al., 2018) includes many time stamps. Every time stamp contains around 7,600 authors with more than 10,000 edges. We took 10 time stamps (i.e., $t_{\max} = 10$) for the PeerRead dataset. We simulated the treatments at every time stamp for all datasets by using $\tau_i \sim \text{Ber}(\text{sigmoid}(\boldsymbol{w}_\tau^\top \boldsymbol{x}_i^t) + \epsilon_{\tau_i})$, where $\boldsymbol{w}_\tau$ is a vector in which every element follows $\mathcal{U}(0, 1)$ independently, and $\epsilon_{\tau_i}$ is a random Gaussian noise. Datasets are shown in Appendix C.

**Baseline.** Our methods were compared with several methods, which are divided into three categories:

- **No graph and no historical information modeling.** BNN (Johansson et al., 2016), CFR-MMD (Shalit et al., 2017), and CFR-Wass (Shalit et al., 2017) address confounders by minimizing distribution discrepancies, MMD, and Wasserstein distance between control and treated groups, respectively. TARNet (Shalit et al., 2017) has the same model architecture as the CFR but no measures for confounders. Above methods all ignore interference and are designed for static data. We run these methods at each time stamp independently, and average results over all time stamps.

Table 1: Results (mean and standard error over ten repeated executions) of treatment effect estimation. Results in boldface represent the lowest mean error.

| Method | Flickr $\sqrt{\epsilon_{\text{PEHE}}}$ | Flickr $\epsilon_{\text{ATE}}$ | Blog $\sqrt{\epsilon_{\text{PEHE}}}$ | Blog $\epsilon_{\text{ATE}}$ | PeerRead $\sqrt{\epsilon_{\text{PEHE}}}$ | PeerRead $\epsilon_{\text{ATE}}$ |
|---|---|---|---|---|---|---|
| TARNet | 4.65 ± 0.06 | 0.54 ± 0.14 | 17.13 ± 0.91 | 2.29 ± 0.30 | 2.87 ± 0.11 | 0.37 ± 0.08 |
| BNN | 5.05 ± 0.00 | **0.15 ± 0.00** | 23.86 ± 0.00 | 2.53 ± 0.00 | 2.83 ± 0.00 | 0.29 ± 0.00 |
| CFR-MMD | 4.60 ± 0.10 | 0.51 ± 0.12 | 17.27 ± 0.99 | 2.14 ± 0.41 | 2.84 ± 0.10 | 0.35 ± 0.06 |
| CFR-Wass | 4.62 ± 0.06 | 0.49 ± 0.11 | 17.49 ± 0.70 | 2.14 ± 0.34 | 2.88 ± 0.12 | 0.37 ± 0.09 |
| GCN-TE | 4.56 ± 0.06 | 0.43 ± 0.07 | 16.90 ± 0.66 | 1.99 ± 0.24 | 2.60 ± 0.06 | 0.22 ± 0.06 |
| GAT-TE | 4.67 ± 0.06 | 0.45 ± 0.09 | 16.45 ± 0.71 | 1.45 ± 0.35 | 2.67 ± 0.06 | 0.26 ± 0.06 |
| GCN-TE+$A_{\text{Pro}}$ | 4.92 ± 0.09 | 0.59 ± 0.19 | 23.45 ± 0.40 | 3.97 ± 0.25 | 2.67 ± 0.10 | 0.31 ± 0.07 |
| GCN-TE* | 4.53 ± 0.07 | 0.49 ± 0.22 | 15.35 ± 1.03 | 1.94 ± 0.67 | 3.27 ± 0.05 | 0.48 ± 0.16 |
| DNDC | 5.53 ± 0.13 | 1.68 ± 0.37 | 23.79 ± 0.03 | 1.64 ± 0.37 | 3.24 ± 0.23 | 0.82 ± 0.14 |
| NEAT | 5.05 ± 0.00 | 0.15 ± 0.00 | 23.86 ± 0.01 | 2.53 ± 0.00 | 2.84 ± 0.00 | 0.29 ± 0.00 |
| DITE (Proposed) | **3.81 ± 0.07** | 0.25 ± 0.09 | 12.71 ± 0.84 | **0.70 ± 0.20** | 1.75 ± 0.05 | **0.11 ± 0.04** |
| DITE$_{\text{ev}}$ (Proposed) | 4.10 ± 0.11 | 0.23 ± 0.07 | **12.38 ± 0.54** | 1.34 ± 0.45 | **1.71 ± 0.05** | **0.11 ± 0.04** |

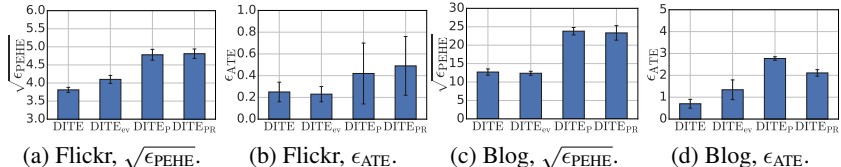

(a) Flickr, $\sqrt{\epsilon_{\text{PEHE}}}$.   (b) Flickr, $\epsilon_{\text{ATE}}$.   (c) Blog, $\sqrt{\epsilon_{\text{PEHE}}}$.   (d) Blog, $\epsilon_{\text{ATE}}$.

Figure 3: Results (mean and standard errors over ten repeated executions) of ablation experiments.

- **No historical information modeling.** The GCN-based method (Ma & Tresp, 2021) captures networked interference on a static graph by GCNs (Welling & Kipf, 2016), and addresses confounders by adding the Hilbert-Schmidt Independence Criterion (HSIC) regularization (Gretton et al., 2005). This method is denoted as GCN-TE. We extend GAT (Veličković et al., 2018) to estimate ITE by replacing GCN with GAT in GCN-TE. This method is denoted as GAT-TE. We run them at each time stamp independently, and average results over all time stamps.
- **Historical information modeling.** DNDC (Ma et al., 2021) models historical information for addressing confounders by GCNs and gated recurrent units (GRUs), but does not model interference. NEAT (Ma et al., 2023) considers how the dynamic network influences the treatment assignment without modeling interference. We extend GCN-TE (Ma & Tresp, 2021) with two schemes: GCN-TE+$A_{\text{Pro}}$ and GCN-TE*. GCN-TE + $A_{\text{Pro}}$ applies GCN-TE to a projection graph $A_{\text{Pro}}$ whose covariates are the concatenation of covariates of previous and the current time stamps, the treatments are the summary of treatments of all time stamps, and the edges in this graph depend on whether there is an edge between two individuals at any time stamp. GCN-TE* learns interference representations at all time stamps and computes the mean representation.

**Metrics.** For all datasets, we calculated $\sqrt{\epsilon_{\text{PEHE}}}$ and $\epsilon_{\text{ATE}}$ to evaluate the error on ITE and ATE (defined as the average effect of a treatment for a group of individuals) estimation, respectively. $\sqrt{\epsilon_{\text{PEHE}}}$ and $\epsilon_{\text{ATE}}$ are defined as follows:

$$\sqrt{\epsilon_{\text{PEHE}}} = \sqrt{\frac{1}{t_{\max}}\frac{1}{n_{\text{test}}}\sum_{t=1}^{t_{\max}}\sum_{i=1}^{n_{\text{test}}}(\Delta_i^t - \hat{\Delta}_i^t)^2}, \quad \epsilon_{\text{ATE}} = \frac{1}{t_{\max}}\sum_{t=1}^{t_{\max}}\left|\frac{1}{n_{\text{test}}}\sum_{i=1}^{n_{\text{test}}}\Delta_i^t - \frac{1}{n_{\text{test}}}\sum_{i=1}^{n_{\text{test}}}\hat{\Delta}_i^t\right|,$$

(8)

where $n_{\text{test}}$ is the size of the test set. We randomly split all datasets into training/validation/test splits with a ratio of 70%/15%/15%. Moreover, details for hyperparameters are introduced in Appendix E.

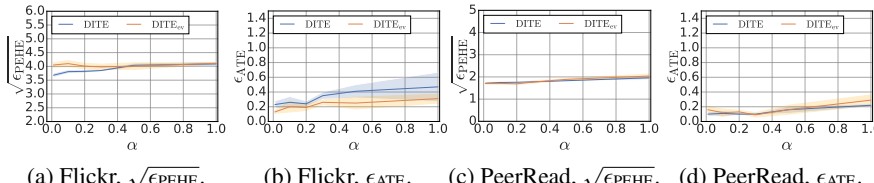

(a) Flickr, $\sqrt{\epsilon_{\text{PEHE}}}$. (b) Flickr, $\epsilon_{\text{ATE}}$. (c) PeerRead, $\sqrt{\epsilon_{\text{PEHE}}}$. (d) PeerRead, $\epsilon_{\text{ATE}}$.

Figure 4: Performance (mean and standard errors over five repeated executions) changes with different values of $\alpha$ on the Flickr and PeerRead datasets.

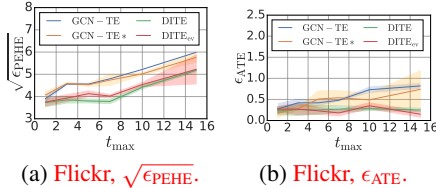

(a) Flickr, $\sqrt{\epsilon_{\text{PEHE}}}$. (b) Flickr, $\epsilon_{\text{ATE}}$.

Figure 5: Performance (mean and standard errors over five repeated executions) changes with different values of $t_{\max}$ on the Flickr dataset.

## 5.2 EXPERIMENTAL RESULTS

**Performance of treatment effect estimation.** To investigate the answer to RQ 1, we conducted experiments and compared the proposed methods with the baseline methods on all datasets. Table 1 lists the results of all methods for ITE and ATE estimation on the test sets of all datasets. It can be seen that the proposed DITE and DITE$_{\text{ev}}$ outperform all baseline methods in ITE estimation with significant performance gaps. In ATE estimation, the proposed methods also outperform all baseline methods on the Blog and PeerRead datasets and achieve a close performance as BNN, which gets the best performance in ATE estimation on the Flickr dataset. These results reveal the powerful ability of the proposed methods to capture dynamic interference.

**Ablation experiments.** To investigate the answer to RQ 2, we conducted ablation experiments. There are two variants of the proposed methods: DITE$_{\text{P}}$ and DITE$_{\text{PR}}$. DITE$_{\text{P}}$ applies DITE to the projection graph $A_{\text{Pro}}$, and DITE$_{\text{PR}}$ replaces the DIM layers with general GCN layers and applies it to the projection graph $A_{\text{Pro}}$ (as detailed in the description for the baseline method GCN-TE+$A_{\text{Pro}}$). Results of ablation experiments are shown in Figure 3 and Appendix F for the PeerRead dataset. These results present significant performance gaps between the DITE/DITE$_{\text{ev}}$ and DITE$_{\text{P}}$/DITE$_{\text{PR}}$, which indicate the importance of the historical information and DIM layers.

**Sensitivity analysis.** To investigate the answer to RQ 3, we conducted experiments with different values of $\alpha$ with the range $\{0.01, 0.1, 0.2, 0.3, 0.5, 1.0\}$ on the Flickr and PeerRead datasets. Results are presented in Figure 4. The results show that no significant changes in performance were observed. This indicates that the proposed methods are not particularly sensitive to the value of $\alpha$.

**A closer look at dynamic interference.** To investigate the answer to RQ 4, we conducted experiments with different $t_{\max}$ on the Flickr dataset, which provides a closer look at dynamic interference. A larger $t_{\max}$ results in more severe dynamic interference among individuals than a small $t_{\max}$. The results are shown in Figure 5. Results show that the performance gap between these baseline methods and the proposed methods with $t_{\max} = 1$ is small, as there is only within-time interference, no dynamic interference. With $t_{\max}$ increase, the performance gap in ITE estimation becomes significant, and the performance of proposed methods achieves better than that of baseline methods in ATE estimation. This indicates the efficacy of capturing dynamic interference of the proposed methods.

Results of additional experiments with another outcome simulation are shown in Appendix H.

## 6 CONCLUSION

In this paper, we study an important research issue: ITE estimation with dynamic interference. To address this issue, we proposed novel approaches and conducted extensive experiments to evaluate the efficacy of the proposed methods, where the results validate their powerful ability in ITE estimation with dynamic interference. We discussed limitations and future work in Appendix K.

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

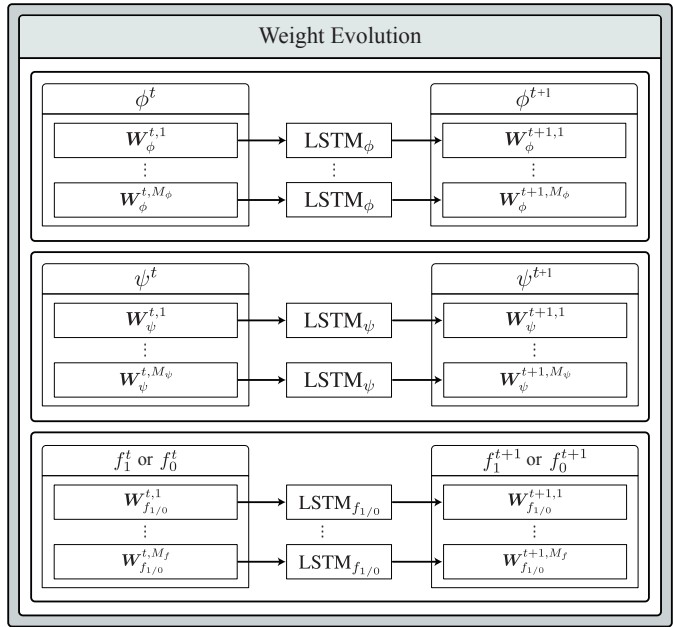

Figure 6: Illustration of weight evolution. In the component of weight evolution, every LSTM takes the weights of a layer at the last time as inputs and generates the weights of a layer at the current time. We use $M_\phi$, $M_\psi$, and $M_f$ LSTMs for $\phi^t$, $\psi^t$ and $f^t_{1/0}$ of DITE$_{\text{ev}}$, respectively. LSTM$_\phi$, LSTM$_\psi$, and LSTM$_{f_{1/0}}$ represent different LSTMs applied to $\phi^t$, $\psi^t$ and $f^t_{1/0}$ of DITE$_{\text{ev}}$, respectively.

## A    IDENTIFIABILITY OF ITE

To identify ITE under interference, a positivity assumption is introduced for neighbor interference in the literature (Jiang & Sun, 2022). However, it is not enough for dynamic interference. We extend the positivity assumption (Jiang & Sun, 2022) to dynamic interference, as follows:

**Assumption 4** *The probability of an individual with dynamic interference to receive treatment or not is always positive,* $0 < P(\tau^t \mid \boldsymbol{x}^t_i, \boldsymbol{s}^t_i) < 1$, $\forall \boldsymbol{x}^t_i, \forall \boldsymbol{s}^t_i$.

This tells the treatment is probabilistic regardless of covariates and dynamic interference received by an individual. Under the set of assumptions, the expected ITE is identifiable, which can be derived as follows:

$$\mathbb{E}[Y^t_i(T^t = 1, S = \boldsymbol{s}^t_i) - Y^t_i(T^t = 0, S = \boldsymbol{s}^t_i) \mid X^t_i = \boldsymbol{x}^t_i]$$

$$= \mathbb{E}[Y^t_i(T^t = 1, S = \boldsymbol{s}^t_i) \mid X^t_i = \boldsymbol{x}^t_i] - \mathbb{E}[Y^t_i(T^t = 0, S = \boldsymbol{s}^t_i) \mid X^t_i = \boldsymbol{x}^t_i]$$

$$\stackrel{(a)}{=} \mathbb{E}[Y^t_i(\tau^t = 1, S = \boldsymbol{s}^t_i) \mid X^t_i = \boldsymbol{x}^t_i, T^t_i = 1, S^t_i = \boldsymbol{s}^t_i] -$$
$$\mathbb{E}[Y^t_i(\tau^t = 0, S = \boldsymbol{s}^t_i) \mid X^t_i = \boldsymbol{x}^t_i, T^t_i = 0, S^t_i = \boldsymbol{s}^t_i]$$

$$\stackrel{(b)}{=} \mathbb{E}[Y^t_i \mid X^t_i = \boldsymbol{x}^t_i, T^t_i = 1, S^t_i = \boldsymbol{s}^t_i] - \mathbb{E}[Y^t_i \mid X^t_i = \boldsymbol{x}^t_i, T^t_i = 0, S^t_i = \boldsymbol{s}^t_i],$$

where (a) is based on the assumption 3, (b) is based on the assumptions 1, 2, and 4. This suggests that once we properly capture $\boldsymbol{s}^t_i$, we can recover the ITE.

## B    ILLUSTRATION OF WEIGHT EVOLUTION

The illustration of weight evolution of DITE$_{\text{ev}}$ is shown in Figure 6.

## C    DETAILS OF DATASETS

Here, the details of each dataset are introduced.

**Flickr dataset (Wang et al., 2013):** Flickr is an online social website where users can share their images.[1] The Flickr dataset (Wang et al., 2013) is collected from this website. In this dataset, each unit is a user of Flickr. There are 7,575 individuals with 229,490˜239,739 undirected edges. At each time stamp, we randomly add/remove edges with the probability of 5%, and perturb covariates at each time stamp using a program provided by literature (Ma et al., 2021). We generated a time-evolving attributed graph and covariates across five different time stamps. Here, we aim to estimate how much recommending a hot photo (treatment) to a user affects the user's experience (outcome) of this photo. In this case, users may share recommended photos with their friends (neighbors), which constitutes networked interference. Furthermore, relationships among users and their interests in social networks often change dynamically. This results in dynamic interference. We used the embeddings of user profiles which are generated by using a list of interest tags of users and provided by literature (Guo et al., 2020) as covariates at the first time stamp. We simulated the treatments at every time stamp as follows:

$$\tau_i \sim \text{Ber}(\text{sigmoid}(\boldsymbol{w}_\tau^\top \boldsymbol{x}_i^t) + \epsilon_{\tau_i}). \tag{9}$$

**BlogCatalog (Li et al., 2015):** BlogCatalog is an online community where users post their blogs.[2] The BlogCatalog (abbreviated as Blog) dataset (Li et al., 2015; 2019; Guo et al., 2020) is collected from the online community. Every node in the graphs is a user of BlogCatalog. There are 5,196 individuals with 161,702˜171,743 undirected edges. At each time stamp, we follow the same process as Flickr to generate a time-evolving attributed graph and covariates across five different time stamps. Here, we aim to estimate how much a recommended blog (treatment) to a user affects the user's experience (outcome) of this blog. In this case, users may share recommended blogs with their friends (neighbors), which constitutes networked interference. Similar to the Flicker, relationships among users and their interests in social networks often change dynamically. This results in dynamic interference. We used the 608-dimensional covariates provided by literature (Guo et al., 2020) for the covariates at the first time stamp. We simulated treatment assignments at every time stamp for the Blog dataset using Eq. equation 9.

**PeerRead (Kang et al., 2018):** PeerRead is a dataset of computer scientific peer reviews for papers, and is used in prior research of ITE estimation (Ma et al., 2021). This dataset contains a real-world dynamic graph of coauthor relations over time. We select 10 time stamps of dynamic graphs which contain 7,601 authors with 11,817˜12,692 undirected edges and 438-dimensional covariates. In this dataset, each node refers to an author, and each edge represents their co-author relationship. The covariates are the bag-of-word representations of their paper titles and abstracts. Treatment is whether the authors' papers contain buzzy words in their titles or abstracts, such as "deep", "neural", "network", and "model". The outcome denotes the citation numbers of authors. We simulated treatment assignments at every time stamp for the PeerRead dataset using Equation (9).

## D    ILLUSTRATION OF OUTCOME SIMULATION

The illustration of outcome simulation for Equation 7 is shown in Figure 7, while the illustration of $f_s$ in Equation 7 is shown in Figure 8.

## E    IMPLEMENTATION

All experiments are conducted with the NVIDIA RTX A6000 GPUs with 48GB GPU memory. For all datasets, we trained the model with the Adam optimizer (Kingma & Ba, 2015). Following existing work for interference (Ma & Tresp, 2021), we consider a transductive setting, i.e., all graph structures $\mathbf{A} = \{\boldsymbol{A}^t\}_{t=1}^{t_{\max}}$, covariates $\mathbf{X} = \{\mathbf{X}^t\}_{t=1}^{t_{\max}}$, and treatments $\mathbf{T} = \{\mathbf{T}^t\}_{t=1}^{t_{\max}}$ were given during the training, validation, and testing phases; whereas only observed outcomes of individuals in the training dataset were provided during training.

We have a simple hyperparameter setting. Unless otherwise specified, we set the learning rate as $0.001$ with a weight decay of $0.001$, the training iterations as $2,000$, the training batch size as $1,024$, $\alpha$ as $0.1$, $M_\phi$ as 3, $M_\psi$ as 2, $M_f$ as 3, and $K$ as 5, $t_{\max}$ as 5, and the dimensions of every layer of $\phi^t$, $\psi^t$, and $f_{1/0}^t$ as 100. We use ReLU (Agarap, 2018) as the activation function for every layer of proposed

---

[1]https://www.flickr.com/
[2]https://www.blogcatalog.com/

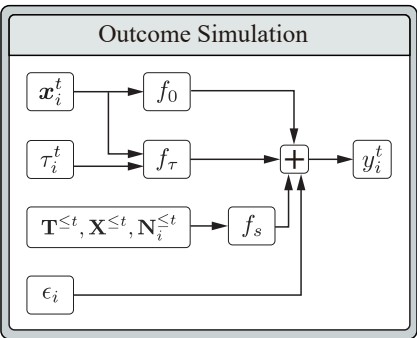

Figure 7: Illustration of outcome simulation for Equation 7.

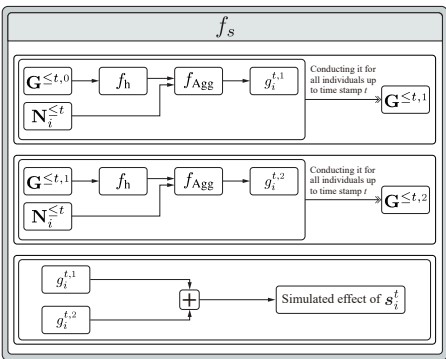

Figure 8: Illustration of $f_s$ in Equation 7.

methods unless otherwise specified. Dropout (Srivastava et al., 2014) and early stopping (Prechelt, 2002) were applied to avoid over-fitting. We used the default or searched hyperparameters within the search range from the corresponding literature for all baseline methods.

## F    ABLATION EXPERIMENTS ON THE PEERREAD DATASET

Results of ablation experiments on the PeerRead dataset are shown in Figure 9.

## G    EXPERIMENTS FOR TRAINING TIME

We conducted experiments to verify the changes in training time with different values of $t_{\max}$ on the Flickr dataset, the results are shown in Table 2. Results show that training time increases linearly with values of $t_{\max}$.

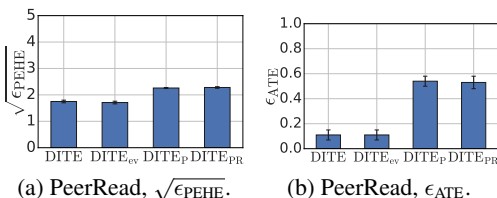

(a) PeerRead, $\sqrt{\epsilon_{\text{PEHE}}}$.        (b) PeerRead, $\epsilon_{\text{ATE}}$.

Figure 9: Results (mean and standard errors over ten repeated executions) of ablation experiments on the PeerRead dataset.

Table 2: Training time (in minutes) with different values of $t_{\max}$ on the Flickr dataset.

| Method | $t_{\max}$ | | | | | |
|---|---|---|---|---|---|---|
| | 1 | 3 | 5 | 7 | 10 | 15 |
| DITE | 6.29 | 18.72 | 31.22 | 45.95 | 62.98 | 107.88 |
| DITE$_{\text{ev}}$ | 6.28 | 19.40 | 32.52 | 49.20 | 69.79 | 104.69 |

Table 3: Training time (in minutes) of the proposed methods on different datasets.

| Method | Flickr | Blog | PeerRead |
|---|---|---|---|
| DITE | 31.22 | 11.57 | 26.08 |
| DITE$_{\text{ev}}$ | 32.52 | 13.16 | 31.61 |

Results of the training time of the proposed methods on different datasets are shown in Table 3. Results show that the training time of the proposed methods is reasonable.

## H  ADDITIONAL EXPERIMENTS WITH OTHER OUTCOME SIMULATION

We use a different $f_s$ to simulate the effect of interference in Equation 7. The new $f_s$ is defined as: $f_s(\mathbf{T}^{\leq t}, \mathbf{X}^{\leq t}, \mathbf{N}_i^{\leq t}) = \sum_{c=1}^{2} g^{t,c}$, where $g^{t,c} = \sum_{t_0=1}^{t} \frac{1}{|\mathbf{N}_i^{t_0}|} \sum_{j \in \mathbf{N}_i^{t_0}} \boldsymbol{w}_j^{t_0,t} g_j^{t-t_0,c-1}$. Different from the original simulation of $f_s$ (in Section 5.1), which first aggregates historical information and then aggregates within-time information, the new simulation of $f_s$ first aggregates within-time information and then aggregates historical information. We call this new simulation as within-time aggregation first (WAF).

### H.1  PERFORMANCE OF TREATMENT EFFECT ESTIMATION FOR WAF SIMULATION.

Table 4 lists the results of all methods for ITE and ATE estimation on the test sets of all datasets for WAF outcome simulation. It can be seen that the proposed DITE and DITE$_{\text{ev}}$ outperform all baseline methods in ITE estimation. Meanwhile, the proposed methods outperform or obtain close performance to baseline methods in ATE estimation. These results reveal again the powerful ability of the proposed methods to capture dynamic interference.

### H.2  ABLATION EXPERIMENTS FOR WAF SIMULATION.

Ablation experiments were also conducted on the Flickr and Blog datasets for WAF simulation. Results are shown in Figure 10. For WAF simulation, there are clear performance gaps between the DITE/DITE$_{\text{ev}}$ and DITE$_{\text{P}}$/DITE$_{\text{PR}}$, which also indicate the importance of the historical information and DIM layers.

### H.3  SENSITIVITY ANALYSIS FOR WAF SIMULATION.

To investigate the sensitivity of proposed methods to $\alpha$ with WAF simulation, we conducted experiments with different values of $\alpha$. Results are shown in Figure 11. These results show that no significant changes in performance were observed, which indicates that the proposed methods are not particularly sensitive to the value of $\alpha$ with WAF simulation.

## I  SUMMARY OF TECHNICAL CONTRIBUTION

To overcome the new and challenging issues: Dynamic interference modeling and ITE estimation under dynamic interference, we proposed new methods. We make necessary modifications, extensions, and combinations of existing methods. We summarize our technical contribution as follows:

Table 4: Results (mean and standard error over ten repeated executions) of treatment effect estimation for WAF simulation. Results in boldface represent the lowest mean error. Results in bold represent the lowest mean error. The proposed DITE and DITE$_{ev}$ outperform all baseline methods in ITE estimation with significant performance gaps. Meanwhile, the proposed methods outperform or are comparable to baseline methods in ATE estimation.

| Method | Flickr | | Blog | | PeerRead | |
|---|---|---|---|---|---|---|
| | $\sqrt{\epsilon_{PEHE}}$ | $\epsilon_{ATE}$ | $\sqrt{\epsilon_{PEHE}}$ | $\epsilon_{ATE}$ | $\sqrt{\epsilon_{PEHE}}$ | $\epsilon_{ATE}$ |
| TARNet | 4.51 ± 0.02 | **0.26 ± 0.08** | 27.44 ± 1.88 | 1.25 ± 0.29 | 3.20 ± 0.07 | 0.27 ± 0.06 |
| BNN | 4.93 ± 0.00 | 0.59 ± 0.00 | 38.28 ± 0.00 | 3.05 ± 0.00 | 3.90 ± 0.00 | 0.48 ± 0.00 |
| CFR-MMD | 4.48 ± 0.02 | 0.29 ± 0.13 | 26.98 ± 2.02 | 1.30 ± 0.35 | 3.22 ± 0.06 | 0.21 ± 0.07 |
| CFR-Wass | 4.49 ± 0.03 | 0.28 ± 0.08 | 27.17 ± 3.08 | 1.11 ± 0.41 | 3.22 ± 0.08 | 0.23 ± 0.07 |
| GCN-TE | 4.54 ± 0.03 | 0.30 ± 0.06 | 25.54 ± 2.12 | 1.25 ± 0.31 | 3.21 ± 0.06 | 0.29 ± 0.06 |
| GAT-TE | 4.55 ± 0.05 | 0.35 ± 0.01 | 24.22 ± 1.91 | 1.03 ± 0.35 | 3.24 ± 0.06 | 0.17 ± 0.05 |
| GCN-TE+$A_{Pro}$ | 4.77 ± 0.05 | 0.82 ± 0.13 | 30.59 ± 2.00 | 2.87 ± 0.39 | 3.25 ± 0.05 | 0.31 ± 0.07 |
| GCN-TE* | 4.35 ± 0.01 | 0.40 ± 0.28 | 24.21 ± 1.20 | **0.82 ± 0.41** | 3.87 ± 0.04 | 0.18 ± 0.14 |
| DNDC | 5.12 ± 0.17 | 1.36 ± 0.60 | 38.16 ± 0.00 | 0.97 ± 0.20 | 3.51 ± 0.04 | 0.22 ± 0.05 |
| NEAT | 4.93 ± 0.00 | 0.58 ± 0.00 | 38.28 ± 0.00 | 3.06 ± 0.00 | 3.89 ± 0.01 | 0.48 ± 0.00 |
| DITE (Proposed) | **3.95 ± 0.08** | 0.43 ± 0.10 | **16.97 ± 1.34** | 0.98 ± 0.42 | 1.97 ± 0.07 | **0.16 ± 0.07** |
| DITE$_{ev}$ (Proposed) | **3.95 ± 0.27** | 0.31 ± 0.09 | 18.61 ± 1.33 | 0.90 ± 0.33 | **1.92 ± 0.07** | 0.17 ± 0.05 |

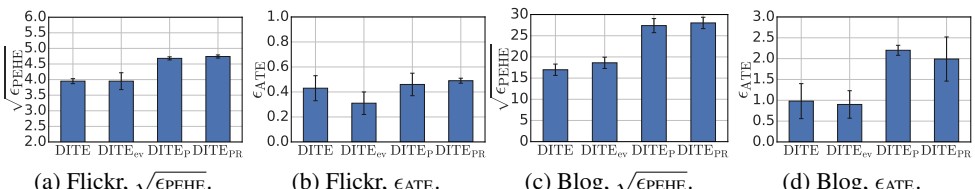

(a) Flickr, $\sqrt{\epsilon_{PEHE}}$.  (b) Flickr, $\epsilon_{ATE}$.  (c) Blog, $\sqrt{\epsilon_{PEHE}}$.  (d) Blog, $\epsilon_{ATE}$.

Figure 10: Results (mean and standard error over ten repeated executions) of ablation experiments for WAF simulation. These results present significant performance gaps between the DITE/DITE$_{ev}$ and DITE$_P$/DITE$_{PR}$.

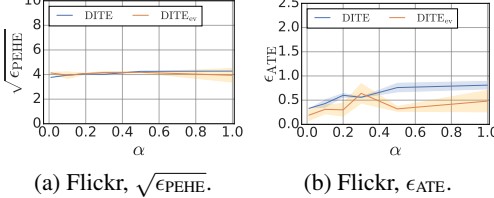

(a) Flickr, $\sqrt{\epsilon_{PEHE}}$.  (b) Flickr, $\epsilon_{ATE}$.

Figure 11: Performance (mean error over five repeated executions) changes of ITE estimation with different $\alpha$ on the Flickr dataset for WAF simulation. Results show that there are no significant performance of the proposed methods changes with different values of $\alpha$.

- DIM Layer for capturing dynamic interference: To address the challenging issue of capturing dynamic interference, we design a new layer named DIM layer to properly model the propagation of dynamic interference among individuals in dynamic graphs. DIM layers have the potential to be applied to other tasks in the causal area with dynamic interference.
- T-attention with a sliding window for capturing cross-time interference: To estimate the importance of cross-time interference from different time stamps, we proposed a weighting network: T-attention with a sliding window. This T-attention mechanism is applied to historical aggregation.
- DITE framework for ITE estimation under dynamic interference: To estimate ITE under dynamic interference, we propose a new framework named DITE.
- $\text{DITE}_{ev}$ for capturing graph dynamics: To properly capture the dynamics of graphs, we propose a variant of DITE, which can dynamically evolve the parameters of the model over time.

## J    DISCUSSION

In our experimental setting, we consider a transductive setting for properly modeling dynamic interference, i.e., information on all networks and individuals is given during training, which follows the existing study (Ma & Tresp, 2021). We now discuss the scenario when complete information on the dynamic networks and covariates of all individuals is not given. As historical information ($< t$) is typically available at a time stamp $t$, this complex scenario can be simplified to one when the information on the network and individuals is partially accessible at the time stamp $t$. In this case, we can mask the covariates of unobserved individuals and hide the edges connecting to these unobserved individuals in the graphs. Here, the performance of the model usually decreases and depends on how much information about individuals and the network is provided.

To properly model dynamic interference, the WA mechanism of the proposed methods is applied to aggregate information from all neighbors of individuals. This might be time-consuming when the input graphs are large. To take this issue into account, we also provide two strategies that can accelerate the WA mechanism, as follows:

- Splitting the graph into some subgraphs. Then, training our methods on each subgraph.
- Applying some sampling mechanisms, such as the neighbor sampling mechanism of Graph-SAGE (Hamilton et al., 2017), to the WA mechanism.

However, there is usually a trade-off between performance and efficiency. Performance of the model may decrease when applying such accelerating technologies.

## K    LIMITATION

For simplicity and focus on capturing dynamic interference, we use a strong unconfoundedness assumption, i.e., Assumption 3 for confounders. This assumption holds when there are no hidden confounders. In other words, all confounders need to be observed from covariates. However, there might exist hidden confounders, which is another popular topic for ITE estimation (Guo et al., 2020; Ma et al., 2021). Thus, if there exist hidden confounders, the component of addressing confounders of DITE/DITE$_{ev}$ can only capture the observed confounders, and omit hidden confounders, which may result in bias in ITE/ATE estimation. Many existing works (Chu et al., 2021; Guo et al., 2020; Ma et al., 2021) present that the hidden confounders can be recovered from the graph and covariates by GNNs. In our proposed component of DIM layers, we use HA mechanism to capture historical information and WA mechanism consisting of GNNs to aggregate information within graphs and minimize MMD of the representations generated by DIM layers to balance the representations. This suggests that our model may be able to capture hidden confounders that are omitted by the addressing confounders by our DIM layers. To investigate this, a deep exploration is needed, which can be achieved by building a bridge between interference and hidden confounders with comprehensive experiments and theoretical analysis. This introduces a potential direction for the future work of this study. Moreover, considering the joint existence of both hidden confounders and dynamic interference is also an interesting topic for future research. However, it is a complex and challenging issue that also requires further exploration.

If we want to estimate ITE at a future time stamp, DITE$_{ev}$ can generate the new weights of the model for the future time stamp. Therefore, DITE$_{ev}$ might be able to estimate ITE in a future time

stamp when the covariates, treatments, and the graph structure at the future time stamp are given. Investigating the performance of this setting can also be the future work of this study. However, a more complex situation is ITE estimation at the future time stamp without any information at the future time stamp. In this case, the proposed methods cannot work, as DITE and DITE$_{ev}$ need covariates, treatments, and graph structures as input. To take this issue into account, we discuss two ideas for extending current DITE$_{ev}$, as follows:

- Training several additional RNNs to generate networks and covariates over different time stamps. During training phase, we can apply a reconstruction loss (Malhotra et al., 2016; Laptev et al., 2018). Let $\hat{A}^t$ be the output of the RNNs for networks and $\hat{x}_i^t$ be the output of the RNNs for covariates. To train these RNNs, we can minimize reconstruction losses $\|\hat{A}^t - A^t\|_2^2$ and $\|\hat{x}_i^t - x_i^t\|_2^2$. These additional RNNs should be trained separately from DITE$_{ev}$. Training them jointly might be challenging and time-consuming, as the dimension of the network is typically high and depends on the number of individuals. However, this idea has an obvious disadvantage: Training these models separately rather than jointly in an end-to-end manner often leads to non-optimal parameters for the downstream task.
- To overcome the disadvantage mentioned in the previous idea, we can incorporate RNNs into the current DITE$_{ev}$ to generate representations of interference and covariates, which is inspired by the existing work (Malhotra et al., 2016; Laptev et al., 2018). Let $\hat{s}_i^t$ be the output of the RNNs for interference representation and $\hat{u}_i^t$ be the output of the RNNs for the representation of covariates. To train these RNNs, we can minimize the reconstruction losses $\|\hat{s}_i^t - s_i^t\|_2^2$ and $\|\hat{u}_i^t - u_i^t\|_2^2$. This idea allows us to jointly train these RNNs and DITE$_{ev}$.

Estimating ITE at future time stamps in the presence of dynamic interference, without any future information, is an interesting but very challenging task. This requires further consideration of the theoretical analyses, and experimental setup to verify these two ideas for overcoming this challenge. This could be considered as a new topic for future research. In addition, these two ideas could be considered as baselines for future research, which aims to estimate ITE at future time stamps without any future information.

