# OpenReview forum: "Dynamic Interference Modeling For Estimating Treatment Effects From Dynamic Graphs"
_ICLR.cc/2025/Conference — Submitted to ICLR 2025_

### Official Review · Reviewer_PQXk · 2024-11-01

**Soundness:** 3
**Presentation:** 2
**Contribution:** 3
**Rating:** 6
**Confidence:** 3

**Summary:**

This paper focuses on the problem of estimating treatment effects from observational graph data in dynamic environments. It proposes the DITE method and its variants to address challenges posed by dynamic interference. Experiments on multiple datasets validate the advantage of the proposed method over existing methods in ITE estimation.
The main contributions of this paper include:

Formalizing the problem of ITE estimation with dynamic interference for the first time, clarifying how individual interference in dynamic environments impacts treatment effect estimation, and providing a clear framework for future research.

Proposing a Dynamic Interference Modeling (DIM) layer that effectively captures dynamic interference through a Historical Aggregation mechanism (HA) and a Within-time Aggregation mechanism (WA). The HA mechanism utilizes a temporal attention mechanism (T-Attention) to estimate the importance of different timestamps and aggregates historical information with weighted importance. The WA mechanism aggregates information within each timestamp using a single-layer GNN. By stacking DIM layers, the model can capture multi-hop dynamic interference among neighbors.

Specifically, to address confounding factors, the approach reduces confounding bias by minimizing the Maximum Mean Discrepancy (MMD) between the covariate representation distributions of the control and treatment groups.

Additionally, the authors design an outcome prediction module based on interference representation, covariate representation, and a treatment training predictor, optimized with a specific loss function. They also propose a variant of DITE that leverages LSTM to evolve model weights, allowing adaptation to dynamic data changes.

**Strengths:**

1、This paper systematically investigates the problem of treatment effect estimation under dynamic interference for the first time, proposing the Dynamic Interference Modeling (DIM) layer to capture dynamic interference between individuals. This is a significant extension of traditional treatment effect estimation research. Notably, the introduction of a temporal attention mechanism (T-Attention) and stacking of single-layer GNNs to handle dynamically changing information offers significant innovation in addressing interference issues in dynamic graph data, providing new directions and insights for this field.

2、The use of Long Short-Term Memory (LSTM) networks to evolve model parameters is unique, enabling the model to better adapt to the dynamic changes in data over time. Compared to models with traditional static parameter settings, this approach performs better when handling data with temporal dynamics.

3、In the problem definition section, relevant concepts and assumptions, such as interference representation, the definition of ITE, and the identifiability proof of ITE under certain assumptions, are clearly articulated, ensuring a rigorous theoretical framework for the study.

4、The design logic of the proposed DITE method and its variants is clear, with each component (such as the DIM layer, confounding factor handling module, and outcome prediction module) having a distinct function and working collaboratively. This setup effectively addresses dynamic interference and estimates treatment effects. For instance, the HA and WA mechanisms within the DIM layer address the key challenges of dynamic interference modeling from the perspectives of historical information aggregation and within-time information aggregation, respectively.

5、In presenting technical methods, experimental setup, and result analysis, the paper uses concise language supported by numerous diagrams (such as architecture diagrams and experiment result graphs). For example, when explaining the working principle of the DIM layer, each step is clearly described through a combination of formulas and text.

**Weaknesses:**

1、To simplify the problem and focus on dynamic interference modeling, the paper adopts a strong no-confounding assumption (Assumption 3), which assumes that all confounders can be observed from covariates. However, in real-world applications, hidden confounders may not be accounted for, potentially leading to bias in ITE/ATE estimation. For example, in complex social or medical scenarios, there may be unobserved variables that simultaneously affect treatment assignment and outcomes.

2、Although the paper acknowledges that hidden confounders are an important issue, it does not delve into whether the proposed methods (e.g., the DIM layer) can effectively capture hidden confounders or address estimation bias that may arise from them. This represents a notable limitation in the study.

3、The paper does not thoroughly explore how to reasonably estimate the ITE at future timestamps in the absence of future information and does not propose a solution to address this scenario.

**Questions:**

1、The paper adopts a strong no-confounding assumption; however, hidden confounders may exist in practice, potentially affecting the accuracy of ITE/ATE estimation. Although the text mentions that existing work could leverage GNNs to recover hidden confounders from the graph and covariates, what potential does the DIM layer have for capturing hidden confounders? Are there plans to conduct experiments or theoretical analyses to evaluate the DIM layer’s effectiveness in handling hidden confounders? Adding experimental studies to address hidden confounders could enhance the evaluation, such as simulating data scenarios with hidden confounders, comparing DITE performance with and without consideration of hidden confounders, and analyzing the DIM layer’s sensitivity and handling capability for hidden confounders, thereby providing a more comprehensive assessment of the method's effectiveness in complex real-world situations.

2、The DITE method and its variants rely on future information to estimate ITE at future timestamps, which may not be available in real-world applications, thus limiting the practicality of the approach. What are the authors’ thoughts on estimating future timestamp ITE without future information? Are there plans to explore new methods or improvements to adapt to this scenario? Investigating how to use historical data and the current model state to predict future timestamp covariates, treatments, and graph structure could support future timestamp ITE estimation. For example, exploring time-series prediction techniques or generative models to create possible future input data, combined with the DITE method, may enable ITE estimation.

---

> ### Author Response · Authors · 2024-11-18
> **Response to Reviewer PQXk**
>
> Thank you for your constructive comments.  Below are our responses to your concerns.
> ### **Summary of Question 1 and Weaknesses 1 and 2**: Limitation of the proposed method when there exist hidden confounders. ###
> **Response**: We had discussed the situation that there exist hidden confounders and some existing works for addressing hidden confounders (such as Deconfounder (Guo et al.,  2020)) in the Limitations section of the Appendices. However, the issue of hidden confounders is not the primary focus of our study, which aims to address the new challenging issues: Dynamic interference modeling and ITE estimation under dynamic interference. Considering the complex scenario involving both hidden confounders and dynamic interference might divert readers' attention from our main objectives. Thus, we borrow the assumption that all confounders are observed in covariates, which is made by many studies for ITE estimation under interference, such as GCN-TE (Ma and Tresp, 2021) and HyperSci (Ma et al., 2022b).
>
>
> To take your comments into account, we conducted experiments to verify the effectiveness of the proposed method in handling hidden confounders.  We use the Flickr$_{k_2=1.0}$ dataset (with a parameter $k_2=1.0$), which is provided by the existing work (Deconfounder, Guo et al.,  2020) for addressing hidden confounders. Importantly, there exist hidden confounders in this dataset. Results are averaged over five executions, as follows:
>
> |  Methods  |  $\sqrt{\epsilon_{\rm{PEHE}}}$   |  $\epsilon_{\rm{ATE}}$  |
> |:----:|:----:|:----:|
> | TARNet | 11.50 ± 0.07 | 2.43 ± 0.30  |
> | CFR-MMD  | 11.42 ± 0.09 | 2.58 ± 0.51  |
> | CFR-Wass |11.45 ± 0.10  | 2.58 ± 0.69  |
> | Deconfounder | 6.08 ± 0.19  | 1.31 ± 0.48  |
> | DITE| 6.23 ± 0.19  | 0.75 ± 0.70  |
>
> Results show that DITE achieves performance comparable to Deconfounder on the  Flickr$_{k_2=1.0}$ dataset, which contains hidden confounders. This implies that the proposed DIM layers may have the potential ability to address hidden confounders when such hidden confounders can be captured from the network. This is an interesting phenomenon, which deserves further exploration by building a bridge between networked interference and hidden confounders in the network. Such exploration requires more comprehensive theoretical analysis and additional experiments, which are beyond the scope of our study, which focuses on modeling dynamic interference and estimating ITE under dynamic interference. Therefore, this can be considered as future work.
>
>
> Moreover, we believe that considering the joint existence of hidden confounders and dynamic interference is a very interesting topic for future research. However, it is a complex and challenging issue that also deserves further exploration.
>
> **Action**:  We will discuss more about this in the part of limitation and future work of the updated version.

---

> > ### Author Response · Authors · 2024-11-22
> >
> > Based on your constructive comments, we have revised and uploaded a new version of our paper.  In the new version, we detailed and highlighted (in **red**) explanations for your concerns in Appendix K.

---

> ### Author Response · Authors · 2024-11-18
> **Response to Reviewer PQXk #2**
>
> ### **Summary of Question 2 and Weaknesses 3**: Concerns about the limitation of the proposed method in ITE estimation at future time and authors’ thoughts on estimating future timestamp ITE without future information. ###
> **Response**: Based on your constructive comments, we discuss two ideas for the issue of ITE estimation without future information. Thank you for the inspiration to explore time-series prediction techniques or generative models to build new models for ITE estimation at future time stamps without using future information. We believe your comment is very helpful for exploring new methods and enhancing the practicality of ITE estimation. We consider the following two ideas:
>
> 1. **Train several additional RNNs to generate networks and covariates over different time stamps**. During training, we can apply a reconstruction loss (such as L2 reconstruction loss) between the output of the RNNs and the networks (or covariates). These additional RNNs should be trained separately from the ${\rm{DITE}}_{\rm{ev}}$ model. Training them jointly might be challenging and time-consuming, as the network’s dimension is typically high and depends on the number of individuals. However, this idea has an obvious disadvantage: Training these models separately rather than jointly in an end-to-end manner often leads to non-optimal parameters for the downstream task.
>
> 2. To overcome the disadvantage mentioned in the previous idea, **we can incorporate RNNs into the current DITE$_{\rm{ev}}$ to generate interference representations and representations of covariates**. In this case, we can apply a reconstruction loss between the interference representation generated by DIM layers and the outputs of the new RNNs during training. A similar strategy can also be applied to the RNNs that generate representations of covariates. This approach allows us to jointly train the new RNNs and ${\rm{DITE}}_{\rm{ev}}$ while avoiding directly generating the network and covariates.
>
> Estimating ITE at future time stamps in the presence of dynamic interference without any future information is an interesting but very challenging task. We will consider more to improve this idea. This needs more consideration about the theoretical analyses, and experimental setup to verify these two ideas for overcoming this challenge. This can be considered as a new topic for future research.
>
>
> **Action**:  We will discuss more about this in the future work of the updated version.

---

### Official Review · Reviewer_Kwsb · 2024-11-03

**Soundness:** 2
**Presentation:** 3
**Contribution:** 2
**Rating:** 5
**Confidence:** 3

**Summary:**

This paper presents a method for estimating Individual Treatment Effects (ITE) in dynamic social networks, where user connections and influences evolve over time. The proposed Dynamic Interference Modeling (DIM) approach captures both within-time and cross-time influences. The architecture, DITE, combines graph neural networks and a time-weighting mechanism to aggregate historical interactions. A variant, DITEev, further adapts parameters over time using LSTM. Experimental results on synthetic and real-world datasets, such as Flickr and BlogCatalog, demonstrate that DITE and DITEev outperform conventional methods.

**Strengths:**

1. The paper studied an interesting and important problem by generalizing the ITE estimation with network interference from static to dynamic network.
2. The description of the proposed model is clear and the model structure of combining time-attention and GNN  is natural and well justified for the proposed problem. Moreover, the introduction of DITEev introduced a natural way to reduce the number of trainable parameters.
3. The authors carry out experiments on three synthetic dataset with comparison to several causal inference baselines.

**Weaknesses:**

1. Assumption 1 implies dependency on the treatment of other users at the same time stamp, but the proposed model does not incorporate this information as input, potentially limiting its alignment with the stated assumptions. Moreover, if the model follows the assumption with the treatment as input, it is not clear how the model can be used to in uplift decisions.
2. The model, even with help of DITEev, introduces a large number of trainbale parameters. The authors should clarify the size of the training dataset used and provide experimental results on how model accuracy depends on the amount of training data.
3. While the proposed algorithm is well-suited for the problem, its real-world applicability is limited. Accessing complete network dynamics and covariates for all individuals is often infeasible, and the algorithm’s runtime on large networks may also pose practical challenges.
4. The experiments are conducted on synthetic datasets, where the ITE aligns with the model assumptions. To validate the model's utility, evaluation on real-world data would be beneficial.
5. Although the authors use Maximum Mean Discrepancy (MMD) to address covariate balance, it remains unclear how the method significantly differs from a non-causal, purely supervised learning setup. Further clarification on this distinction is needed.

**Questions:**

Please see weakness section.

---

> ### Author Response · Authors · 2024-11-18
> **Response to Reviewer Kwsb**
>
> Thank you for your constructive comments.  Below are our responses to your concerns.
> ## Response to Weaknesses:
> ### **Weakness 1.1**: Assumption 1 implies dependency on the treatment of other users at the same time stamp, but the proposed model does not incorporate this information as input, potentially limiting its alignment with the stated assumptions. ###
> **Response**: The inputs of $\psi^t$ of our proposed method are $\mathbf{X}^{\leq t}$, $\mathbf{T}^{\leq t}$, and $\mathbf{A}^{\leq t}$ (as detailed in Line 276), where $\mathbf{T}^{\leq t}$ (the description of superscript $\leq t$ is detailed in Line 161 for the original version or Line 165 for the updated version) contains historical treatment assignments and treatments assigned to other individuals at the time stamp $t$.
>
>
>
> ### **Weakness 1.2**: Moreover, if the model follows the assumption with the treatment as input, it is not clear how the model can be used to in uplift decisions. ###
> **Response**:  We use $f^t_1(\boldsymbol{s}_i^t,\boldsymbol{u}_i^t)- f^t_0(\boldsymbol{s}_i^t,\boldsymbol{u}_i^t)$ to estimate ITE (as detailed in Lines 362 and 363), where ${f^t_1}$  predicts the potential outcome with treatment = 1 and $f^t_0$ predicts the potential outcome with treatment = 0. To model interference, we take treatments as inputs, which follow existing works for ITE estimation under interference, such as GCN-TE (Ma and Tresp, 2021) and HyperSci  (Ma et al., 2022b). Moreover, information on observed treatments is also used to train $f^t_1$, and $f^t_0$. If the observed treatment = 1 for a unit at time $t$, the outcome of this unit will be used to train $f^t_1$;  If the observed treatment = 0 for a unit, the outcome of this unit will be used to train $f^t_0$, as illustrated in Figure 2(a), detailed in Section 4.3 and Equation (6).
>
> We can estimate ITE at time $t$ and past time stamps (< t) by using our ITE estimator, which can help us to understand the change of individuals, as well as the change of their treatment effect. This can help us to deeply analyze the effect of the treatment, which results in more reasonable decisions.
>
>
> **Action**: To take your comment into account, we will add more descriptions for our ITE estimator in the updated version.
>
>
> ### **Weakness 2**:  The authors should clarify the size of the training dataset used and provide experimental results on how model accuracy depends on the amount of training data. ###
> **Response**: We had clarified the ratio of the training set (as detailed in Lines 480 and 481 for the original version of the paper or Line 485 for the updated version) and the size of every dataset (as detailed in the part of Dataset description and Appendix C). Specifically, we use the ratio of 70% data (as detailed in Lines 480 and 481 for the original version of the paper or Line 485 for the updated version)  to train all methods.

---

> ### Author Response · Authors · 2024-11-18
> **Response to Reviewer Kwsb #2**
>
> ### **Weakness 3.1**: Its real-world applicability is limited. Accessing complete network dynamics and covariates for all individuals is often infeasible. ###
> **Response**:
> As historical information ($<t$) is typically available at time stamp $t$, we can simplify the situation you mentioned to one that discusses whether the information of some individuals on the network is accessible at time $t$.
> Many existing studies allow using complete information on the network and covariates of all individuals to train their models, such as GCN (Welling and Kipf, 2016), Deconfounders (Guo et al.,  2020), and GCN-TE (Ma and Tresp, 2021). However, their applicability is not limited. For example, GCN (Welling and Kipf, 2016) had been widely used in many applications and areas. Following the above works, we make the information on covariates and network accessible at time $t$ for properly addressing the new challenging issues: Dynamic interference modeling and ITE estimation under dynamic interference.  Now, we provide a strategy to take your comment into account. When some parts of information on covariates and networks are missing, we can mask the covariates of unobserved individuals and hide the edges connecting to these unobserved individuals in the graphs. In this case, the performance of the model usually decreases and depends on how much information about individuals and the network is provided.
>
>
> **Action**: We will discuss the situation when complete network dynamics and covariates of all individuals are not given in the appendices of the updated version.
>
>
> ### **Weakness 3.2**: The algorithm’s runtime on large networks may also pose practical challenges. ###
> **Response**:
>  There are many existing works for accelerating GNN, such as GraphSAGE, GraphSAINT, and FastGCN, which can be used to reduce computation in the within-time aggregation of proposed methods.  Importantly, there usually be a trade-off in performance and efficiency.   We aim to address the new issues: Dynamic interference modeling and ITE estimation under dynamic interference. Thus, we prioritize performance in ITE estimation under dynamic interference. When readers want to further accelerate our methods, we recommend that they extend our methods by combining our methods with these existing works for accelerating GNN.
>
> To take your comment into account, we discuss two strategies that can accelerate the within-time aggregation part of our methods in training.
>
> 1. **Splitting the graph into some sub-graphs when the input graph is large.** Then,  training our methods on each subgraph.
>
> 2. **Applying some neighbor sampling mechanisms** (e.g., the neighbor sampling mechanism of GraphSAGE) to the within-time aggregation mechanism of our method (as detailed in equation (4)).
>
> As described above, the proposed methods are extendable. We also provide the training time (in minutes) of our methods on corresponding datasets:
>
> | Methods |Flickr| Blog  |  PeerRead |
> |:--:|:--:|:--:|:--:|
> | DITE | 31.22  |  11.57 | 26.08 |
> | DITE$_{\rm{ev}}$ |  32.52 |  13.16 | 31.61 |
>
> The training time of our method is reasonable, even without deliberate acceleration of the within-time aggregation component. If the strategies mentioned above are applied to speed up this part, it may further improve training efficiency.
>
> **Action**: We will discuss the situation when readers want to accelerate our methods and provide the Table of training time of our methods in the Appendices of the updated version.
>
>
> ### **Weakness 4**: The experiments are conducted on synthetic datasets. To validate the model's utility, evaluation on real-world data would be beneficial. ###
> **Response**: Our experiments are conducted on semi-synthetic datasets, which is a common choice in most existing works for ITE estimation in the causal area, such as Deconfounder (Guo et al.,  2020), GCN-TE (Ma and Tresp, 2021), HyperSci (Ma et al., 2022b), and DNDC (Ma et al., 2021). The ground truth of ITE is hard to collect, as we cannot get the counterfactual outcomes from real-world data. This is a common choice to use semi-synthetic datasets, as conducting RCT to collect the ground truth of ITE is time-consuming and expensive. Thus, using semi-synthetic datasets to conduct experiments is reasonable for research studies in the causal area.
>
>
>
> ### **Weakness 5**: It remains unclear how the method significantly differs from a non-causal, purely supervised learning setup. ###
> **Response**: **ITE estimation is an important task in causal inference**, and different from non-causal learning setup that predicts outcomes only. We had introduced the concept and role of ITE estimation in Section 1 with an example of its application (as detailed in Lines 38 ~ 44), as well as the details of problem setting in Section 3. For more details on the difference between ITE estimation and supervised learning setup, please refer to the survey (Yao et al., 2021).

---

> ### Author Response · Authors · 2024-11-22
>
> Based on your constructive comments, we have revised and uploaded a new version of our paper.  In the new version, we detailed and highlighted (in **red**) explanations for your concerns in Sections 4.1 and 4.3, Table 3,  Appendices G and J.

---

### Official Review · Reviewer_k7s7 · 2024-11-04

**Soundness:** 2
**Presentation:** 2
**Contribution:** 2
**Rating:** 5
**Confidence:** 3

**Summary:**

This paper addresses the estimation of individual treatment effects (ITE) in the presence of dynamic interference, where individual outcomes are influenced not only by current treatments but also by past interactions and treatments from previous neighbors within a dynamic network context. The authors propose a novel architecture that aggregates historical information from individuals and their neighbors using graph neural networks, incorporating a weighting mechanism to assess the importance of different timestamps. Additionally, they introduce a variant of their method that adapts model parameters over time using long short-term memory (LSTM) networks. Experiments on multiple datasets demonstrate the superiority of their approach in capturing dynamic interference compared to existing methods. The paper is well-written and thoroughly investigates the estimation of individual treatment effects (ITE) in dynamic networks. The authors empirically validate their model, emphasizing its significance in understanding the effects of social dynamics on treatment outcomes.

**Strengths:**

1. The paper fills a significant gap in the literature by formalizing the problem of ITE estimation with dynamic interference. The introduction of the DIM layer represents a novel contribution that distinguishes this study from existing methods.
2. The DITE framework is well-structured to address challenges related to historical information aggregation, within-time interference, and cross-time interference. This comprehensive approach is crucial for advancing the field of treatment effect estimation in dynamic contexts.
3. The authors provide extensive experimental results that demonstrate the superiority of their proposed methods over existing techniques. This empirical evidence enhances the credibility of their claims and underscores the practical relevance of their work.

**Weaknesses:**

1. The background and introduction sections do not sufficiently explain ITE, particularly in the context of dynamic networks. A more detailed introduction would help readers appreciate the research value in this area.
2. The overall framework diagram for the experiments lacks clarity. While the mechanism diagrams for DITE and DIM are present, a comprehensive experimental framework diagram is missing. This oversight makes it difficult to quickly grasp the main contributions and optimization mechanisms of the article. I suggest including a clearer framework diagram to facilitate understanding for readers.
3. In the results section, the main experimental results include only two comparison metrics. I recommend adding more widely used metrics to better demonstrate the effectiveness of the experimental results. Additionally, conducting further experiments to validate the model's performance—beyond simple comparisons of standard deviation and mean—would be beneficial. Presenting these comparisons in more intuitive charts would also help to better showcase the experimental outcomes.

**Questions:**

See above

---

> ### Author Response · Authors · 2024-11-18
> **Response to Reviewer k7s7**
>
> Thank you for your constructive comments on the presentation.  Below are our responses to your concerns.
> ## Response to Weaknesses:
> ### **Weakness 1**: The background and introduction sections do not sufficiently explain ITE, particularly in the context of dynamic networks. ###
>
> **Response**:
>  We had introduced the role and basic concept of ITE estimation (quantifies the relative
> change of an individual outcome with/without treatment, as detailed in Lines 43 and 44) by using a commerce case in the first paragraph (detailed in Lines 38~44) of the section of Introduction (i.e., Section 1 in our manuscript). Furthermore, in the second paragraph of the section of Introduction, we introduced that outcomes and information of individuals and networks can change over time in the dynamic graphs, which implies ITE can also change over time. To take your comment into account, we will add some descriptions in Section 1 to emphasize it.
>
> **Action**: We will add some descriptions to emphasize the dynamics of ITE in the updated version.
>
> ###  **Weakness 2**: The overall framework diagram for the experiments lacks clarity. ###
>
> **Response**:
> We had introduced the details of the experimental setting (as detailed in Section 5.1). Reviewers EDpc and PQXk point out that the details of experimental settings had already been introduced in the Strengths of their comments.   Importantly, the outcome simulation had been introduced detailedly in Equation (7).   Considering the outcome simulation is complex and your constructive comment, we will make a diagram for the outcome simulation. Thank you for your constructive comment on the presentation of experiments.
>
> **Action**: We will add a diagram for the outcome simulation in the updated version.
>
> ###  **Weakness 3.1**: the main experimental results include only two comparison metrics. Presenting these comparisons in more intuitive charts would also help to better showcase the experimental outcomes. ###
> **Response**:
> The metrics used in our paper are widely used in causal inference, there are many existing works that consider these two metrics, such as CFR (Shalit et al., 2017), HyperSci (Ma et al., 2022b), HINITE (Lin et al., 2023), and DNDC (Ma et al., 2021). Thus, they are enough and reasonable for verifying the performance of our methods.
>
>
> ###  **Weakness 3.2**:  Presenting these comparisons in more intuitive charts would also help to better showcase the experimental outcomes. ###
> **Response**: We had used histograms to show the results of ablation experiments (as shown in Figure 3) and used line charts to show the results of different hyperparameters (as shown in Figures 4 and 5). Importantly, we clearly highlighted the results with the lowest mean error of those of all methods on the corresponding dataset **in boldface** in Table 1.
> Since Table 1 includes ten methods, three datasets, and two metrics, it is challenging to use intuitive charts (e.g., histograms) to represent these results effectively. A small figure would make the results difficult to interpret, while a large figure would occupy excessive space. Separating the results in Table 1 into six large figures would also require a significant amount of space.

---

> ### Author Response · Authors · 2024-11-22
>
> Based on your constructive comments, we have revised and uploaded a new version of our paper.  In the new version, we detailed and highlighted (in **red**) explanations for your concerns in Section 1, the part of the description for outcome simulation, Appendix D, and Figures 7 and 8.

---

### Official Review · Reviewer_EDpc · 2024-11-08

**Soundness:** 3
**Presentation:** 3
**Contribution:** 3
**Rating:** 5
**Confidence:** 4

**Summary:**

This work is designed to learn a dynamic interference modeling for estimating treatment effects. Genearlly speaking, the proposed problem is interesting. Prior research has focused on static interference in fixed networks, but real-world networks often change over time, leading to dynamic interference. To tackle the issues in this field, this study introduces a model for dynamic interference, aggregating both past and current information of individuals. Specifically, the proposed model uses a mechanism for summarizing historical data, graph neural networks for time-stamped interactions, and an attention-based weighting system to learn the contribution of each timestamp. Additionally, it adjusts parameters gradually to reflect evolving information dynamics, leveraging long short-term memory (LSTM) to update them. Experiments show that this approach outperforms existing models, highlighting the significance of accounting for dynamic interference in ITE estimation.

**Strengths:**

1. The authors evaluate various degraded version of the proposed model and tasks in real-world datasets to show the effectiveness of the main components, clearly detailing which components from previous research they incorporate and build upon.
2. The proposed problem is interesting.
3. The paper is well-presented and easy to follow.
4. The details of experimental settings are introduced such as the simulation methods.

**Weaknesses:**

1. The technical contribution of this work is limited. Several works focus on solving the dynamic interference in ITE estimation, so the proposed problem is not new. For example, "Treatment Effect Estimation Amidst Dynamic Network Interference in Online Gaming Experiments" also learns the dynamic interference. The model design seems combine many existing approaches. Stacking dnn layer to capture dynamic information and using attention mechanism to automatically learn the contribution of each timestamp are quite general methods.
2. More advanced gnn-based models should be considered such as lightgcn, gat.
3. Some important references are missing such as Treatment Effect Estimation Amidst Dynamic Network Interference in Online Gaming Experiments.
4. In a closer look at dynamic interference part, a larger t_max should be considered. It can be better to describe the effect of the historical information. More historical information, better performance. Also, showing the trade-off between the running time and the value of t_max may make the results more convincing.

**Questions:**

1. It is better to introduce more technical contributions of the proposed model.
2. more gnn-based models should be compared.

---

> ### Author Response · Authors · 2024-11-18
> **Response to Reviewer EDpc**
>
> Thank you for your constructive comments. Below are our responses to your concerns.
> ## Response to Weaknesses and Questions:
> ###  **Weakness 1.1**: Several works (such as [R1]) focus on solving the dynamic interference in ITE estimation, so the proposed problem is not new. ###
> **Response**:  After carefully reviewing the recommended paper [R1], we found that it tackles a different problem setting from ours. We summarize the differences as follows:
>
> [R1] focuses on the ATE estimation for online games with dynamic networks only, but **static covariates and treatments**. In contrast, our methods focus on ITE estimation on dynamic graph data that have **not only dynamic networks but also dynamic covariates and treatments**. Our study introduces a new and more complex challenge:  **Cross-time interference**.  Cross-time interference can occur between two individuals who are **not neighboring** at any time stamp, as shown in Figure 1 and detailed in Section 1 (Lines 81 ~ 83 in the original version or Lines 83 ~ 86 in the updated version). This is a novel and significant challenge.  Without properly modeling it, we may end up with wrong ITE estimation, which can result in wrong decision. Moreover, [R1] considers a setting of A/B testing, whereas we consider estimating ITE from observational data, which is inherently more challenging.
>
>
> **Action**: We will discuss the differences between our study and [R1] in the updated version.
>
> ###  **Weaknesses 1.2 and 3 and Question 1**: Concerns about the technical contribution.  ###
>
> **Response**: *Proposing solutions to a new and challenging issue constitutes a significant technical contribution*. To overcome the challenges of dynamic interference modeling and ITE estimation under dynamic interference, we make necessary modifications, extensions, and combinations of existing methods. We summarize our technical contributions as follows:
> 1. **DIM Layer for capturing dynamic interference**: To address the challenging issue of capturing dynamic interference, we design a new layer: the **DIM layer** (detailed in Section 4.1). Unlike traditional GNN layers, the proposed DIM layer applies a historical aggregation mechanism before every within-time aggregation (i.e., a single-layered GNN). This step is crucial for capturing cross-time interference. By stacking DIM layers, we are able to model dynamic interference properly. In contrast, traditional GNN methods for dynamic graphs typically use multi-layered GNNs to learn node embeddings at each time stamp and then combine the learned embeddings from all time stamps at once, which can fail to capture cross-time interference properly. For example, results in Table 1 show that the baseline GCN-TE*, which combines the outputs of the final GNN layer at different times only after completing the multiple GNN layers across all time stamps, obtains high errors in ITE estimation. In contrast, our method achieves a lower error in ITE estimation. *It is promising to extend DIM layers for other tasks involving dynamic interference in the causal area.*
>
> 2. **T-attention with a sliding window for capturing cross-time interference**: To estimate the importance of cross-time interference from different time stamps, we proposed a weighting network: **T-attention**. This T-attention mechanism is applied to historical aggregation (as detailed in Lines 290 ~ 297 in the original version or Lines 289 ~ 293 in the updated version). To take the scalability issue into account, we apply a sliding window (as detailed in Lines 320 ~ 331 in the original version of the paper or Lines 316 ~ 323 in the updated version)  to T-attention and historical information aggregation. This enables the individual to focus on the interference from the most recent time stamps, which often make the most significant contribution to cross-time interference.
>
> 3. **DITE framework for ITE estimation under dynamic interference**: To estimate ITE under dynamic interference, we propose a new framework: **DITE** (detailed in Section 4).
>
> 4. **DITE$_{\rm{ev}}$ for capturing graph dynamics**: To properly capture the dynamics of graphs, we also propose **DITE$_{\rm{ev}}$** (as detailed in Section 4.4), which dynamically evolves the parameters of the model over time by using LSTMs.
>
> **Action**: We will discuss the technical contributions in the updated version of our manuscript.

---

> ### Author Response · Authors · 2024-11-18
> **Response to Reviewer EDpc #2**
>
> ###  **Weakness 2 and Question 2**: More advanced gnn-based models should be considered. ###
> **Response**: Although there had been many GNN-based methods compared in our experiments, such as GCN-TE, GCN-TE*, and DNDC, we added two GNN-based methods to take your comment into account, as follows:
> 1. GAT-TE. As recommended by the comment, we extend GAT to ITE estimation by replacing the GCN of GCN-TE with GAT [R2].
>
> 2. NEAT [R3] is designed to estimate ITE under a dynamic environment. However, it only considers how the dynamic network influences the treatment assignment but does not model interference.
>
> The results of GAT-TE and NEAT [R3] are as follows:
>
> On the Flickr dataset.
>
> |  Methods  |  $\sqrt{\epsilon_{\rm{PEHE}}}$   |  $\epsilon_{\rm{ATE}}$  |
> |:----:|:----:|:----:|
> | GAT-TE | 4.67 ± 0.06 | 0.45 ± 0.09  |
> | NEAT  | 5.05 ± 0.00 | 0.15 ± 0.00  |
> | DITE |3.81 ± 0.07  | 0.25 ± 0.09  |
> | DITE$_{\rm{ev}}$ | 4.10 ± 0.11  | 0.23 ± 0.07  |
>
> On the Blog dataset.
>
> |  Methods  |  $\sqrt{\epsilon_{\rm{PEHE}}}$   |  $\epsilon_{\rm{ATE}}$  |
> |:----:|:----:|:----:|
> | GAT-TE | 16.45 ± 0.71 | 1.45 ± 0.35  |
> | NEAT  | 23.86 ± 0.01 | 2.53 ± 0.00  |
> | DITE |12.71 ± 0.84  | 0.70 ± 0.20  |
> | DITE$_{\rm{ev}}$ | 12.38 ± 0.54  | 1.34 ± 0.45  |
>
> On the PeerRead dataset.
>
> |  Methods  |  $\sqrt{\epsilon_{\rm{PEHE}}}$   |  $\epsilon_{\rm{ATE}}$  |
> |:----:|:----:|:----:|
> | GAT-TE | 2.67 ± 0.06 | 0.26 ± 0.06  |
> | NEAT  | 2.84 ± 0.00 | 0.29 ± 0.00  |
> | DITE |1.75 ± 0.05  | 0.11 ± 0.04  |
> | DITE$_{\rm{ev}}$ | 1.71 ± 0.05  | 0.11 ± 0.04  |
>
> **Action**:  We will add these two new baselines in the updated version of our manuscript.
>
> ###  **Weakness  4.1**: In a closer look at dynamic interference part, a larger t_max should be considered. ###
> **Response**:  Although many experiments had been conducted with different values of $t_{\rm{max}}$ (the size of all time stamps), which reveal the changes in the performance gap between the proposed methods and baselines with different values of $t_{\rm{max}}$, we considered a larger $t_{\rm{max}} = 15$ and conducted experiments on the Flickr dataset to take your comment into account. The results are shown as follows:
>
> |  Methods  |  $\sqrt{\epsilon_{\rm{PEHE}}}$   |  $\epsilon_{\rm{ATE}}$  |
> |:--------:|:--------:|:--------:|
> | GCN-TE |  5.99 ± 0.03 |  0.82 ± 0.08  |
> | GCN-TE*  | 5.76 ± 0.22 | 0.74 ± 0.44 |
> | DITE |  5.18 ± 0.06|  0.25 ± 0.04  |
> | DITE$_{\rm{ev}}$ | 5.22 ± 0.65   | 0.15 ± 0.07   |
>
> **Action**: We will add these results in the updated version.
> ###  **Weakness  4.2**: Showing the running time of different values of $t_{\rm{max}}$. ###
> **Response**: The training time (in minutes) of different values of $t_{\rm{max}}$ for DITE are shown as follows:
>
>
> | Methods |$$t_{\rm{max}}=1$$|  $$t_{\rm{max}}=3$$  |  $$t_{\rm{max}}=5$$ |$$t_{\rm{max}}=7$$  |$$t_{\rm{max}}=10$$  |$$t_{\rm{max}}=15$$  |
> |:--:|:--:|:--:|:--:|:--:|:--:|:--:|
> | DITE | 6.29  | 18.72   |  31.22| 45.95  |62.98  |  107.88  |
> | DITE$_{\rm{ev}}$ |   6.28 | 19.40   | 32.52  |  49.20   | 69.79   | 104.69   |
>
> Results show that the training time increases linearly with $t_{\rm{max}}$.
>
> **Action**: We will add these results in the updated version.
>
>
>
>
> ## Reference
> [R1] Treatment Effect Estimation Amidst Dynamic Network Interference in Online Gaming Experiments.
>
> [R2] Graph Attention Networks. ICLR 2018.
>
> [R3] A Look into Causal Effects under Entangled Treatment in Graphs: Investigating the Impact of Contact on MRSA Infection. KDD 2023.

---

> ### Author Response · Authors · 2024-11-22
>
> Based on your constructive comments, we have revised and uploaded a new version of our paper.  In the new version, we detailed and highlighted (in **red**) explanations for your concerns in Section 2, the part of the description for the baseline, Tables 1, 2, and 4, Figure 5, and Appendices G and I.

---

### Author Response · Authors · 2024-11-18
**To all reviewers and ACs.**

We sincerely thank the ACs and reviewers for their invaluable time, effort, and constructive comments in evaluating our work. Your insightful comments have greatly contributed to improving the quality of our research. In our responses, **Response** outlines our answers to the reviewers' comments, while **Action** details the specific revisions we plan to implement in the updated version of the manuscript during the camera-ready phase.

---

> ### Author Response · Authors · 2024-11-22
> **A new version of paper is uploaded**
>
> We uploaded the revised version of our manuscript based on reviewers' constructive comments,  where we highlight the main changes in **red**. We believe the revision has strengthened the manuscript.
>
> Thank you for your time and consideration!

---

### Meta-Review · Area_Chair_nbMT · 2024-12-23

**Metareview:**

The paper tackles individual treatment effects (ITE) in the presence of dynamic interference, where individual outcomes are influenced not only by current treatments but also by past interactions and treatments from previous neighbors within a dynamic network. All the reviewers highlighted that the problem is interesting and the presentation of the proposed framework is clear. However, they all have a number of concerns that, in terms of overall score, led them to either be mildly negative (3 out of 4 reviewers) or mildly positive (one out of 1 reviewers). The main concerns were regarding the significance of the technical contribution, the assumptions and practicality of the method, and the experiments using semi-synthetic data. As a consequence, I am unable to recommend acceptance.

**Additional Comments On Reviewer Discussion:**

The authors put a significant effort in addressing the concerns raised by the reviewers. However, the reviewers were not persuaded by the rebuttal to follow-up or change their overall recommendation.

---

### Decision · Program_Chairs · 2025-01-22

Reject